# Offline Reinforcement Learning with Value-Based Episodic Memory

**Xiaoteng Ma**[1][*] **Yiqin Yang**[1][*] **Hao Hu**[2][*] **Qihan Liu**[1],
**Jun Yang**[1][†] **Chongjie Zhang**[2][†] **Qianchuan Zhao**[1], **Bin Liang**[1]
[1]Department of Automation, Tsinghua University
[2]Institute for Interdisciplinary Information Sciences, Tsinghua University
`{ma-xt17,yangyiqi19,hu-h19,lqh20}@mails.tsinghua.edu.edu`
`{yangjun603,chongjie,zhaoqc,bliang}@tsinghua.edu.cn`

## ABSTRACT

Offline reinforcement learning (RL) shows promise of applying RL to real-world problems by effectively utilizing previously collected data. Most existing offline RL algorithms use regularization or constraints to suppress extrapolation error for actions outside the dataset. In this paper, we adopt a different framework, which learns the $V$-function instead of the $Q$-function to naturally keep the learning procedure within the offline dataset. To enable effective generalization while maintaining proper conservatism in offline learning, we propose Expectile $V$-Learning (EVL), which smoothly interpolates between the optimal value learning and behavior cloning. Further, we introduce implicit planning along offline trajectories to enhance learned $V$-values and accelerate convergence. Together, we present a new offline method called Value-based Episodic Memory (VEM). We provide theoretical analysis for the convergence properties of our proposed VEM method, and empirical results in the D4RL benchmark show that our method achieves superior performance in most tasks, particularly in sparse-reward tasks. Our code is public online at https://github.com/YiqinYang/VEM.

## 1 INTRODUCTION

Despite the great success of deep reinforcement learning (RL) in various domains, most current algorithms rely on interactions with the environment to learn through trial and error. In real-world problems, particularly in risky and safety-crucial scenarios, interactions with the environment can be expensive and unsafe, and only offline collected datasets are available, such as the expert demonstration or previously logged data. This growing demand has led to the emergence of *offline reinforcement learning* (offline RL) to conduct RL in a supervised manner.

The main challenge of offline RL comes from the actions out of the dataset's support (Kumar et al., 2019; 2020). The evaluation of these actions that do not appear in the dataset relies on the generalization of the value network, which may exhibit *extrapolation error* (Fujimoto et al., 2019). This error can be magnified through bootstrapping, leading to severe estimation errors. A rapidly developing line of recent work (Fujimoto et al., 2019; Kumar et al., 2020; Ghasemipour et al., 2021; Yang et al., 2021) utilizes various methods to constrain optimistic estimation on unseen actions, such as restricting available actions with a learned behavior model (Fujimoto et al., 2019) or penalizing the unseen actions with additional regularization (Kumar et al., 2020). However, confining learning *within the distribution of the dataset* can be insufficient for reducing extrapolation errors.

Another line of methods, on the contrary, uses the returns of the behavior policy as the signal for policy learning, as adopted in Wang et al. (2018); Peng et al. (2019); Chen et al. (2020). By doing so, they keep the value learning procedure completely *within the dataset*. However, the behavior policy of the dataset can be imperfect and insufficient to guide policy learning. To achieve a trade-off between imitation learning and optimal value learning while confines learning *within the dataset*,

---

[*]Equal contribution. Listing order is random.
[†]Equal advising.

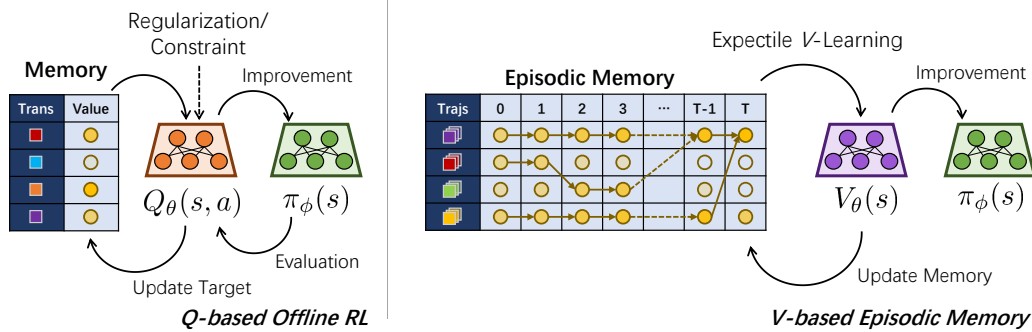

Figure 1: The diagram of algorithms. The left side denotes the general $Q$-based offline RL methods. The right side is the framework of our proposed approach (VEM). Q-based methods learns bootstrapped $Q$-values, but requires additional constraint or penalty for actions out of the dataset. Our method, on the contrary, learns bootstrapped $V$-values while being completely confined within the dataset without any regularization.

we propose Expectile $V$-learning (EVL), which is based on a new expectile operator that smoothly interpolates between the Bellman expectation operator and optimality operator.

To better solve long-horizon and sparse-reward tasks, we further propose using value-based planning to improve the advantage estimation for policy learning. We adopt an implicit memory-based planning scheme that strictly plans within offline trajectories to compute the advantages effectively, as proposed in recent advances in episodic memory-based methods (Hu et al., 2021). Together, we present our novel framework for offline RL, Value-based Episodic Memory (VEM), which uses expectile $V$-learning to approximate the optimal value with offline data and conduct implicit memory-based planning to further enhance advantage estimation. With the properly learned advantage function, VEM trains the policy network in a simple regression manner. We demonstrate our algorithm in Figure 1, and a formal description of our algorithm is provided in Algorithm 1.

The contributions of this paper are threefold. First, we present a new offline $V$-learning method, EVL, and a novel offline RL framework, VEM. EVL learns the value function through the trade-offs between imitation learning and optimal value learning. VEM uses a memory-based planning scheme to enhance advantage estimation and conduct policy learning in a regression manner. Second, we theoretically analyze our proposed algorithm's convergence properties and the trade-off between contraction rate, fixed-point bias, and variance. Specifically, we show that VEM is provably convergent and enjoys a low concentration rate with a small fixed-point bias. Finally, we evaluate our method in the offline RL benchmark D4RL (Fu et al., 2020). Comparing with other baselines, VEM achieves superior performance, especially in the sparse reward tasks like AntMaze and Adroit. The ablation study shows that VEM yields accurate value estimates and is robust to extrapolation errors.

## 2 BACKGROUND

**Preliminaries**. We consider a Markov Decision Process (MDP) $M$ defined by a tuple $(\mathcal{S}, \mathcal{A}, P, r, \gamma)$, where $S$ is the state space, $A$ is the action space, $P(\cdot \mid s, a) : \mathcal{S} \times \mathcal{A} \times \mathcal{S} \to \mathbb{R}$ is the transition distribution function, $r(s, a) : \mathcal{S} \times \mathcal{A} \to \mathbb{R}$ is the reward function and $\gamma \in [0, 1)$ is the discount factor. We say an environment is deterministic if $P(s' \mid s, a) = \delta(s' = f(s, a))$ for some deterministic transition function $f$, where $\delta(\cdot)$ is the Dirac function. The goal of an RL agent is to learn a policy $\pi : \mathcal{S} \times \mathcal{A} \to \mathbb{R}$, which maximizes the expectation of a discounted cumulative reward: $\mathcal{J}(\pi) = \mathbb{E}_{s_0 \sim \rho_0, a_t \sim \pi(\cdot \mid s_t), s_{t+1} \sim P(\cdot \mid s_t, a_t)} \left[ \sum_{t=0}^{\infty} \gamma^t r(s_t, a_t) \right]$, where $\rho_0$ is the distribution of the initial states.

**Value-based Offline Reinforcement Learning Methods**. Current offline RL methods can be roughly divided into two categories according to types of learned value function: $Q$-based and $V$-based methods. $Q$-based methods, such as BCQ (Fujimoto et al., 2019), learn $Q$-function for policy learning and avoid selecting unfamiliar actions via constraints or penalty. On the contrary, $V$-based methods (Peng et al., 2019; Siegel et al., 2020; Chen et al., 2020) learns the value of behavior policy $V^\mu(s)$ with the trajectories in the offline dataset $\mathcal{D}$ and update policy as a regression problem. Based on the learned $V$-function, $V$-based methods like AWR (Peng et al., 2019) updates the policy using advantage-weighted regression, where each state-action pair is weighted according

to the exponentiated advantage:

$$\max_{\phi} \mathcal{J}_{\pi}(\phi) = \mathbb{E}_{(s_t, a_t) \sim \mathcal{D}} \left[ \log \pi_{\phi}(a_t \mid s_t) \exp \left( R_t - V^{\mu}(s_t) \right) \right]. \tag{1}$$

**Episodic Memory-Based Methods**. Inspired by psychobiology, episodic memory-based methods store experiences in a non-parametric table to fast retrieve past successful strategies when encountering similar states. Model-free episodic control (Blundell et al., 2016a) updates the memory table by taking the maximum return $R(s, a)$ among all rollouts starting from same state-action pair $(s, a)$. Hu et al. (2021) proposes *Generalizable Episodic Memory*, which extends this idea to the continuous domain, and proposes updating formula with a parametric memory $Q_{\theta}^{EM}$.

## 3 METHOD

In this section, we describe our novel offline method, value-based episodic memory, as depicted in Figure 1. VEM uses expectile $V$-learning (EVL) to learn $V$-functions while confines value learning *within the dataset* to reduce extrapolation error. EVL uses an expectile operator that interpolates between Bellman expectation operator and optimality operator to balance behavior cloning and optimal value learning. Further, VEM integrates memory-based planning to improve the advantage estimation and accelerate the convergence of EVL. Finally, generalized advantage-weighted learning is used for policy learning with enhanced advantage estimation. A formal description for the VEM algorithm is shown in Algorithm 1 in Appendix A.1.

### 3.1 EXPECTILE V-LEARNING

To achieve a balance between behavior cloning and optimal value learning, we consider the Bellman expectile operator defined as follows:

$$((\mathcal{T}_{\tau}^{\mu})V)(s) := \arg\min_{v} \mathbb{E}_{a \sim \mu(\cdot|s)} \left[ \tau [\delta(s,a)]_{+}^{2} + (1-\tau)[\delta(s,a)]_{-}^{2} \right] \tag{2}$$

where $\mu$ is the behavior policy, $\delta(s,a) = \mathbb{E}_{s' \sim P(\cdot|s,a)}[r(s,a) + \gamma V(s') - v]$ is the expected one-step TD error, $[\cdot]_{+} = \max(\cdot, 0)$ and $[\cdot]_{-} = \min(\cdot, 0)$. This operator resembles the *expectile* statistics (Newey & Powell, 1987; Rowland et al., 2019) and hence its name. We can see that when $\tau = 1/2$, this operator is reduced to Bellman expectation operator, while when $\tau \to 1$, this operator approaches Bellman optimality operator, as depicted in Lemma 3.

We use the following toy example to further illustrate the trade-offs achieved by EVL. Consider a random generated MDP. When the operator can be applied exactly, the Bellman optimality operator is sufficient to learn the optimal value $V^{*}$. However, applying operators with an offline dataset raises a noise on the actual operator due to the estimation error with finite and biased data. We simulate this effect by adding random Gaussian noise to the operator. Applying the optimality operator on offline datasets can lead to severe overestimation due to the maximization bias and bootstrapping. The value estimation learned by EVL, on the contrary, achieves a trade-off between learning optimal policy and behavior cloning and can be close to the optimal value with proper chosen $\tau$, as depicted in Figure 2. The noise upon the operator largely depends on the size of the dataset. Estimation error can be significant with insufficent data. In this case, we need a small $\tau$ to be conservative and be close to behavior cloning. When the dataset is large and we are able to have an accurate estimation for the operator,

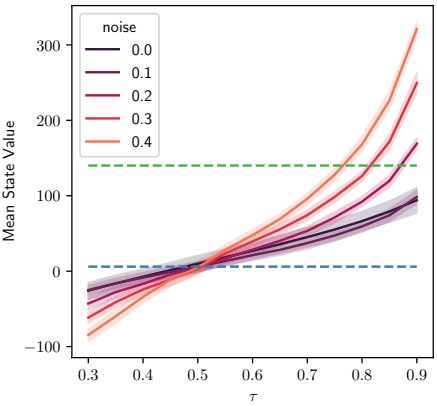

Figure 2: Trade-offs of EVL between generalization and conservatism in a random MDP. The green line shows the optimal value and the blue line shows the value of behavior policy. The curve is averaged over 20 MDPs.

we can use a larger $\tau$ to recover the optimal policy. By adjusting $\tau$, the expectile operator can accommodate variant types of datasets. However, the expectile operator in Equation 2 does not have a

closed-form solution. In practice, we consider the one-step gradient expectile operator

$$((\mathcal{T}_g)_\tau^\mu V)(s) = V(s) + 2\alpha \mathbb{E}_{a \sim \mu(\cdot|s)} \left[ \tau \left[ \delta(s,a) \right]_+ + (1-\tau)[\delta(s,a)]_- \right], \tag{3}$$

where $\alpha$ is the step-size. Please refer to Appendix B.1 for the detailed derivation. For notational convenience, we use $\mathcal{T}_\tau^\mu$ to denote the one-step gradient expectile operator $(\mathcal{T}_g)_\tau^\mu$ hereafter.

We consider the case where the dynamics are nearly-deterministic like robotic applications, and we remove the expectation over the next states in the operator. This leads to a practical algorithm, Expectile $V$-Learning, where we train the value network to minimize the following loss:

$$\mathcal{J}_V(\theta) = \mathbb{E}_{(s,a,s') \sim \mathcal{D}} \left[ \left( \hat{V}(s) - V_\theta(s) \right)^2 \right],$$

$$\hat{V}(s) = V_{\theta'}(s) + 2\alpha \left[ \tau \left[ \delta(s,a,s') \right]_+ + (1-\tau)[\delta(s,a,s')]_- \right], \tag{4}$$

where $\hat{V}$ is the target value after applying one-step gradient expectile operator and $\delta(s,a,s') = r(s,a) + \gamma V_{\theta'}(s') - V_{\theta'}(s)$. $V$-function and the target $\hat{V}$-function are parameterized by $\theta$ and $\theta'$, respectively. EVL is guaranteed to converge with concentration rate $\gamma_\tau = 1 - 2(1-\gamma)\alpha \max\{\tau, 1-\tau\}$. Please refer to Section 4 for a detailed analysis.

## 3.2 IMPLICIT MEMORY-BASED PLANNING

Although EVL reduces the extrapolation error, it is still a challenging problem to bootstrap over long time horizons due to estimation errors with a fixed dataset. Therefore, we propose using value-based planning to conduct bootstrapping more efficiently. We adopt an implicit memory-based planning scheme that strictly plans within offline trajectories to avoid over-optimistic estimations in the planning phase. This is aligned with recent advances in episodic memory-based methods (Hu et al., 2021), but we conduct this planning on expectile $V$-values rather than $Q$-values. Specifically, we compare the best return so far along the trajectory with the value estimates $\hat{V}$ and takes the maximum between them to get the augmented return $\hat{R}_t$:

$$\hat{R}_t = \begin{cases} r_t + \gamma \max(\hat{R}_{t+1}, \hat{V}(s_{t+1})), & \text{if} \quad t < T, \\ r_t, & \text{if} \quad t = T, \end{cases} \tag{5}$$

where $t$ denotes steps along the trajectory, $T$ is the episode length, and $\hat{V}$ is generalized from similar experiences. This procedure is conducted recursively from the last step to the first step along the trajectory, forming an implicit planning scheme within the dataset to aggregate experiences along and across trajectories. Further, the back-propagation process in Equation 5 can be unrolled and rewritten as follows:

$$\hat{R}_t = \max_{0 < n \leq n_{\max}} \hat{V}_{t,n}, \quad \hat{V}_{t,n} = \begin{cases} r_t + \gamma \hat{V}_{t+1,n-1} & \text{if} \quad n > 0, \\ \hat{V}(s_t) & \text{if} \quad n = 0, \end{cases} \tag{6}$$

where $n$ denotes different length of rollout steps and $\hat{V}_{t,n} = 0$ for $n > T$.

## 3.3 GENERALIZED ADVANTAGE-WEIGHTED LEARNING

Based on $\hat{R}_t$ calculated in Section 3.2, we can conduct policy learning in a regression form, as adopted in return-based offline RL methods (Nair et al., 2020; Siegel et al., 2020; Peng et al., 2019):

$$\max_\phi \mathcal{J}_\pi(\phi) = \mathbb{E}_{(s_t,a_t) \sim \mathcal{D}} \left[ \log \pi_\phi(a_t \mid s_t) \cdot f \left( \hat{A}(s_t, a_t) \right) \right], \tag{7}$$

where $\hat{A}(s_t, a_t) = \hat{R}_t - \hat{V}(s_t)$ and $f$ is an increasing, non-negative function. Please refer to Appendix C.1 for the detailed implementation of Equation 7. Note that $\hat{R}_t$ is not the vanilla returns in the dataset, but the enhanced estimation calculated by implicit planning from $\hat{V}_t$, as opposed with other return based methods. Please refer to Algorithm 1 and Section 4 for implementation details and theoretical analysis.

## 4 THEORETICAL ANALYSIS

In this section, we first derive the convergence property of expectile $V$-Learning. Then, we demonstrate that memory-based planning accelerates the convergence of the EVL. Finally, we design a toy example to demonstrate these theoretical analyses empirically. Please refer to Appendix B for the detailed proofs of the following analysis.

### 4.1 CONVERGENCE PROPERTY OF THE EXPECTILE V-LEARNING

In this section, we assume the environment is deterministic. We derive the contraction property of $\mathcal{T}_\tau^\mu$ as the following statement:

**Lemma 1.** *For any $\tau \in (0, 1)$, $\mathcal{T}_\tau^\mu$ is a $\gamma_\tau$-contraction, where $\gamma_\tau = 1 - 2\alpha(1 - \gamma)\min\{\tau, 1 - \tau\}$.*

*Proof.* We introduce two more operators to simplify the analysis:

$$(\mathcal{T}_+^\mu V)(s) = V(s) + \mathbb{E}_{a \sim \mu}[\delta(s, a)]_+, (\mathcal{T}_-^\mu V)(s) = V(s) + \mathbb{E}_{a \sim \mu}[\delta(s, a)]_-. \tag{8}$$

Next we show that both operators are non-expansion (e.g., $\|\mathcal{T}_+^\mu V_1 - \mathcal{T}_+^\mu V_2\|_\infty \leq \|V_1 - V_2\|_\infty$). Finally, we rewrite $\mathcal{T}_\tau^\mu$ based on $\mathcal{T}_+^\mu$ and $\mathcal{T}_-^\mu$ and we prove that $\mathcal{T}_\tau^\mu$ is a $\gamma_\tau$-contraction. Please refer to Appendix B.2 for the complete proof. $\square$

Based on Lemma 1, we give a discussion about the step-size $\alpha$ and the fraction $\tau$:

**About the step-size $\alpha$.** Generally, we always want a larger $\alpha$. However, $\alpha$ must satisfy that $V(s) + 2\alpha\tau\delta(s, a) \leq \max\{r(s, a) + \gamma V(s'), V(s)\}$ and $V(s) + 2\alpha(1 - \tau)\delta(s, a) \geq \min\{r(s, a) + \gamma V(s'), V(s)\}$, otherwise the $V$-value will be overestimated. Thus, we must have $2\alpha\tau \leq 1$ and $2\alpha(1 - \tau) \leq 1$, which infers that $\alpha \leq \frac{1}{2\max\{\tau, 1-\tau\}}$. When $\alpha = \frac{1}{2\max\{\tau, 1-\tau\}}$, we have $\gamma_\tau = 1 - 2\alpha\min\{\tau, 1 - \tau\}(1 - \gamma) = 1 - \frac{\min\{\tau, 1-\tau\}}{\max\{\tau, 1-\tau\}}(1 - \gamma)$.

**About the fraction $\tau$.** It is easy to verify that $\gamma_\tau$ approaches to 1 when $\tau \to 0$ or $\tau \to 1$, which means that with a larger $\tau$ the contractive property is getting weaker. The choice of $\tau$ makes a trade-off between the learning stability and the optimality of values. We further point out that when $\tau = 1$, the Expectile $V$-learning degrades as a special case of the generalized self-imitation learning (Tang, 2020), which losses the contractive property.

Next, we prove that $\mathcal{T}_\tau^\mu$ is monotonous improving with respect to $\tau$:

**Lemma 2.** *For any $\tau, \tau' \in (0, 1)$, if $\tau' \geq \tau$, we have $\mathcal{T}_{\tau'}^\mu V(s) \geq \mathcal{T}_\tau^\mu V(s), \forall s \in S$.*

Based on the Lemma 2, we derive that $V_\tau^*$ is monotonous improving with respect to $\tau$:

**Proposition 1.** *Let $V_\tau^*$ denote the fixed point of $\mathcal{T}_\tau^\mu$. For any $\tau, \tau' \in (0, 1)$, if $\tau' \geq \tau$, we have $V_{\tau'}^*(s) \geq V_\tau^*(s), \forall s \in S$.*

Further, we derive that $V_\tau^*$ gradually approaches $V^*$ with respect to $\tau$:

**Lemma 3.** *Let $V^*$ denote the fixed point of Bellman optimality operator $\mathcal{T}^*$. In the deterministic MDP, we have $\lim_{\tau \to 1} V_\tau^* = V^*$.*

Based on the above analysis, we have the following conclusion:

**Remark 1.** *By choosing a suitable $\tau$, we can achieve the trade-off between the contraction rate and the fixed point bias. Particularly, a larger $\tau$ introduces a smaller fixed point bias between $V_\tau^*$ and $V^*$, and produces a larger contraction rate $\gamma_\tau$ simultaneously.*

### 4.2 VALUE-BASED EPISODIC MEMORY

In this part, we demonstrate that the memory-based planning effectively accelerates the convergence of the EVL. We first define the VEM operator as:

$$(\mathcal{T}_{\text{vem}} V)(s) = \max_{1 \leq n \leq n_{\max}}\{(\mathcal{T}^\mu)^{n-1}\mathcal{T}_\tau^\mu V(s)\}, \tag{9}$$

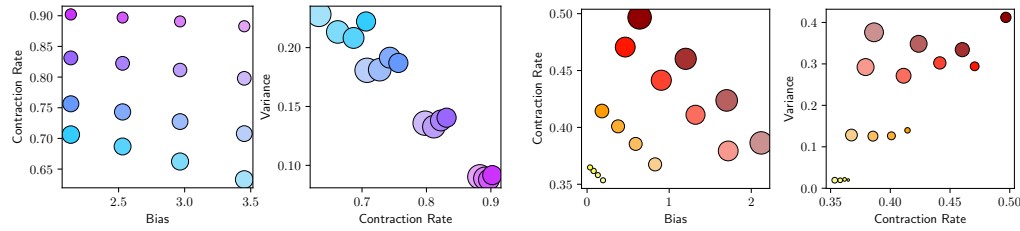

(a) The maximal rollout step $n_{\max}$.          (b) The different behavior policies.

Figure 3: A toy example in the random MDP. In both figures, the color darkens with a larger $\tau$ ($\tau \in \{0.6, 0.7, 0.8, 0.9\}$). The size of the spots is proportional to the relative scale of the third variable: (a) Change $n_{\max}$. From magenta to blue, $n_{\max}$ is set as $1, 2, 3, 4$ in order. (b) Change the behavior polices $\mu$, where $\mu(s) = \text{softmax}(Q^*(s, \cdot)/\alpha)$. From light yellow to dark red, the $\alpha$ is set as $0.1, 0.3, 1, 3$ in order.

where $n_{\max}$ is the maximal rollout step for memory control. Then, we derive that multi-step estimation operator $\mathcal{T}_{\text{vem}}$ does not change the fixed point and contraction property of $\mathcal{T}_{\tau}^{\mu}$:

**Lemma 4.** *Given* $\tau \in (0, 1)$ *and* $n_{\max} \in \mathbb{N}^+$, $\mathcal{T}_{\text{vem}}$ *is a* $\gamma_{\tau}$*-contraction. If* $\tau > \frac{1}{2}$, $\mathcal{T}_{\text{vem}}$ *has the same fixed point as* $\mathcal{T}_{\tau}^{\mu}$.

Next, we derive that the contraction rate of $\mathcal{T}_{\text{vem}}$ depends on the dataset quality. Further, we demonstrate that the convergence rate of $\mathcal{T}_{\text{vem}}$ is quicker than $\mathcal{T}_{\tau}^{\mu}$ even the behavior policy $\mu$ is random:

**Lemma 5.** *When the current value estimates* $V(s)$ *are much lower than the value of behavior policy,* $\mathcal{T}_{\text{vem}}$ *provides an optimistic update. Formally, we have*

$$|\mathcal{T}_{\text{vem}}V(s) - V_{\tau}^*(s)| \le \gamma^{n^*(s)-1}\gamma_{\tau}\|V - V_{n^*,\tau}^{\mu}\|_{\infty} + \|V_{n^*,\tau}^{\mu} - V_{\tau}^*\|_{\infty}, \forall s \in S, \qquad (10)$$

*where* $n^*(s) = \arg\max_{0 < n \le n_{\max}}\{(\mathcal{T}^{\mu})^{n-1}\mathcal{T}_{\tau}^{\mu}V(s)\}$, $V_{n^*,\tau}^{\mu}$ *is the fixed point of* $(\mathcal{T}^{\mu})^{n^*(s)-1}\mathcal{T}_{\tau}^{\mu}$ *and it is the optimal rollout value starting from* $s$.

This lemma demonstrates that $\mathcal{T}_{\text{vem}}$ can provide an optimistic update for pessimistic value estimates. Specifically, the scale of the update depends on the quality of the datasets. If the behavior policy $\mu$ is expert, which means $V_{n^*,\tau}^{\mu}$ is close to $V_{\tau}^*$. Then, following the lemma, the contraction rate will be near to $\gamma^{n^*(s)-1}\gamma_{\tau}$. Moreover, if the initial value estimates are pessimistic (e.g., the initialized value function with zeros), we will have $n^*(s) \approx n_{\max}$, indicating that the value update will be extremely fast towards a lower bound of $V_{\tau}^*$. On the contrary, if $\mu$ is random, we have $n^*(s) \approx 1$ and the value update will be slow towards $V_{\tau}^*$.

**Remark 2.** *By choosing a suitable* $n_{\max}$, *we can achieve the trade-off between the contraction rate and the estimation variance, i.e., a larger* $n_{\max}$ *yields a fast update towards a lower bound of fixed point and tolerable variances empirically. Meanwhile, the choice of* $n_{\max}$ *does not introduce additional bias, and the fixed point bias is totally controlled by* $\tau$.

### 4.3 Toy Example

We design a toy example in the random deterministic MDP to empirically demonstrate the above analysis. Following (Rowland et al., 2020), we adopt three indicators, including update variance, fixed-point bias, and contraction rate, which is shown in Figure 3. Specifically, the contraction rate is $\sup_{V \ne V'} \|\mathcal{T}_{\text{vem}}V - \mathcal{T}_{\text{vem}}V'\|_{\infty}/\|V - V'\|_{\infty}$, the bias is $\|V_{\text{vem}}^* - V^*\|_{\infty}$ and the variance is $\mathbb{E}\left[\|\hat{\mathcal{T}}V - \mathcal{T}_{\text{vem}}V\|_2^2\right]^{\frac{1}{2}}$, where $\hat{\mathcal{T}}_{\text{vem}}$ is the stochastic approximation of $\mathcal{T}_{\text{vem}}$ and $V_{\text{vem}}^*$ is the fixed pointed of $\mathcal{T}_{\text{vem}}$. First, the experimental results in Figure 3(a) demonstrate that the relationship of $n$-step estimation and $\tau$. Formally, the contraction rate decreases as $n$ becomes larger, and the fixed-point bias increases as $\tau$ becomes smaller, which are consistent with Lemma 1 and Lemma 2. Figure 3(a) also shows that the variance is positively correlated with $n$. Second, the experimental results in Figure 3(b) demonstrate that the relationship of dataset quality and $\tau$. The higher dataset quality corresponds to the lower contraction rate and variance, which is consistent with Lemma 5.

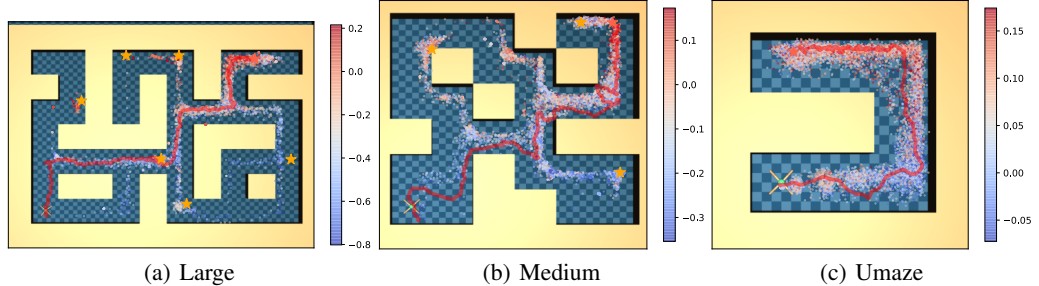

Figure 4: Visualization of the value estimation in various AntMaze tasks. Darker colors correspond to the higher value estimation. Each map has several terminals (golden stars) and one of which is reached by the agent (the light red star). The red line is the trajectory of the ant.

## 5 RELATED WORK

**Offline Reinforcement Learning.** Offline RL methods (Kumar et al., 2019; Siegel et al., 2020; Argenson & Dulac-Arnold, 2020; Wu et al., 2021; Dadashi et al., 2021; Kostrikov et al., 2021; Jin et al., 2021; Rashidinejad et al., 2021) can be roughly divided into policy constraint, pessimistic value estimation, and model-based methods. Policy constraint methods aim to keep the policy to be close to the behavior under a probabilistic distance (Fujimoto et al., 2019; Peng et al., 2019; Nair et al., 2020). Pessimistic value estimation methods like CQL (Kumar et al., 2020) enforces a regularization constraint on the critic loss to penalize overgeneralization. Model-based methods attempt to learn a model from offline data, with minimal modification to the policy learning (Kidambi et al., 2020; Yu et al., 2020; Janner et al., 2019). However, these methods have to introduce additional behavioral policy models, dynamics models, or regularization terms (Zhang et al., 2020b;a; Lee et al., 2021). Another line of methods uses empirical return as the signal for policy learning, which confines learning within the dataset but leads to limited performance (Levine et al., 2020; Geist et al., 2019; Wang et al., 2021).

**Episodic Control.** Episodic control aims to store good past experiences in a non-parametric memory and rapidly latch into past successful policies when encountering similar states instead of waiting for many optimization steps (Blundell et al., 2016b). Pritzel et al. (2017) and Lin et al. (2018) introduce a parametric memory, which enables better generalization through neural networks. Our work is closely related to recent advances in Hu et al. (2021), which adopts an implicit planning scheme to enable episodic memory updates in continuous domains. Our method follows this implicit scheme, but conducts planning with expectile $V$-values to avoid overgeneralization on actions out of dataset support.

## 6 EXPERIMENTS

In our experiments, we aim to answer the following questions: 1) How does our method performe compared to state-of-the-art offline RL algorithms on the D4RL benchmark dataset? 2) How does implicit planning affect the performance on sparse reward tasks? 3) Can expectile $V$-Learning effectively reduces the extrapolation error compared with other offline methods? 4) How does the critical parameter $\tau$ affect the performance of our method?

### 6.1 EVALUATION ENVIRONMENTS

We ran VEM on AntMaze, Adroit, and MuJoCo environments to evaluate its performance on various types of tasks. Precisely, the AntMaze navigation tasks control an 8-DoF quadruped robot to reach a specific or randomly sampled goal in three types of maps. The reward in the AntMaze domain is highly sparse. The Adroit domain involves controlling a 24-DoF simulated hand tasked with hammering a nail, opening a door, twirling a pen, or picking up and moving a ball. On the adroit tasks, these datasets are the following, "human": transitions collected by a human operator,

| Type | Env | VEM(Ours) | VEM($\tau$=0.5) | BAIL | BCQ | CQL | AWR |
|------|-----|-----------|-----------------|------|-----|-----|-----|
| fixed | umaze | **87.5±1.1** | 85.0±1.5 | 62.5 ± 2.3 | 78.9 | 74.0 | 56.0 |
| play | medium | **78.0±3.1** | 71.0±2.5 | 40.0 ± 15.0 | 0.0 | 61.2 | 0.0 |
| play | large | **57.0±5.0** | 45.0±2.5 | 23.0±5.0 | 6.7 | 11.8 | 0.0 |
| diverse | umaze | 78.0 ± 1.1 | 75.0±5.0 | 75.0±1.0 | 55.0 | **84.0** | 70.3 |
| diverse | medium | **77.0±2.2** | 60.0±5.0 | 50.0±10.0 | 0.0 | 53.7 | 0.0 |
| diverse | large | **58.0 ± 2.1** | 48.0±2.7 | 30.0±5.0 | 2.2 | 14.9 | 0.0 |
| human | door | **11.2±4.2** | 6.9±1.1 | 0.0±0.1 | -0.0 | 9.1 | 0.4 |
| human | hammer | **3.6±1.0** | 2.5±1.0 | 0.0±0.1 | 0.5 | 2.1 | 1.2 |
| human | relocate | 1.3±0.2 | 0.0±0.0 | 0.0±0.1 | 0.5 | **2.1** | -0.0 |
| human | pen | 65.0±2.1 | 55.2±3.1 | 32.5±1.5 | **68.9** | 55.8 | 12.3 |
| cloned | door | **3.6±0.3** | 0.0±0.0 | 0.0±0.1 | 0.0 | 3.5 | 0.0 |
| cloned | hammer | 2.7±1.5 | 0.5±0.1 | 0.1±0.1 | 0.4 | **5.7** | 0.4 |
| cloned | pen | **48.7±3.2** | 27.8±2.2 | 46.5±3.5 | 44.0 | 40.3 | 28.0 |
| expert | door | **105.5±0.2** | 104.8±0.2 | 104.7±0.3 | 99.0 | - | 102.9 |
| expert | hammer | **128.3±1.1** | 102.3±5.6 | 123.5±3.1 | 114.9 | - | 39.0 |
| expert | relocate | **109.8±0.2** | 101.0±1.5 | 94.4±2.7 | 41.6 | - | 91.5 |
| expert | pen | 111.7±2.6 | 115.2±1.3 | **126.7±0.3** | 114.9 | - | 111.0 |
| random | walker2d | 6.2±4.7 | 6.2±4.7 | 3.9±2.5 | 4.9 | **7.0** | 1.5 |
| random | hopper | **11.1±1.0** | 10.8±1.2 | 9.8±0.1 | 10.6 | 10.8 | 10.2 |
| random | halfcheetah | 16.4±3.6 | 2.6±2.1 | 0.0±0.1 | 2.2 | **35.4** | 2.5 |
| medium | walker2d | 74.0±1.2 | 16.6±0.1 | 73.0±1.0 | 53.1 | **79.2** | 17.4 |
| medium | hopper | 56.6±2.3 | 56.6±2.3 | **58.2±1.0** | 54.5 | 58.0 | 35.9 |
| medium | halfcheetah | **47.4±0.2** | 45.3±0.2 | 42.6±1.2 | 40.7 | 44.4 | 37.4 |

Table 1: Performance of VEM with four offline RL baselines on the AntMaze, Adroit, and MuJoCo domains with the normalized score metric proposed by D4RL benchmark, averaged over three random seeds with ± standard deviation. Scores range from 0 to 100, where 0 corresponds to a random policy performance, and 100 indicates an expert. We use the results in Fu et al. (2020) for AWR and BCQ, and use the results in Kumar et al. (2020) for CQL. The results of BAIL come from our implementation according to the official code (https://github.com/lanyavik/BAIL).

"cloned": transitions collected by a policy trained with behavioral cloning interacting in the environment + initial demonstrations, "expert": transitions collected by a fine-tuned RL policy interacting in the environment. As for the MuJoCo tasks, the datasets are "random": transitions collected by a random policy, "medium": transitions collected by a policy with suboptimal performance. The complete implementation details are presented in Appendix C.

## 6.2 PERFORMANCE ON D4RL TASKS

As shown in Table 1, VEM achieves state-of-the-art performance on most AntMaze tasks and has a significant improvement over other methods on most Adroit tasks. VEM also achieves good performances in MuJoCo domains. We find that VEM has low value estimation errors in all tasks, which promotes its superior performance. However, as a similar training framework, BAIL only has reasonable performances on simple offline tasks, such as MuJoCo. Please refer to Appendix D.2 for the complete training curves and value estimation error on D4RL.

To further analyze the superior performance of VEM in the sparse reward tasks, we visualize the learned value estimation in AntMaze tasks, which is shown in Figure 4. Experimental results show that VEM has the higher value estimates on the critical place of the map (e.g., corners) since various trajectories in the datasets are connected. The accurate value estimation leads to its success on complex sparse reward tasks.

## 6.3 ANALYSIS OF VALUE ESTIMATION

As both Expectile $V$-Learning (EVL) and Batch Constrained $Q$-Learning (BCQ) (Fujimoto et al., 2019) aim to avoid using the unseen state-action pairs to eliminate the extrapolation error, we replace EVL in VEM with BCQ (named BCQ-EM) to evaluate the effectiveness of the EVL module.

The experimental results in Figure 9 in Appendix D.1 indicate that the performance of BCQ-EM is mediocre, and BCQ reaches performance significantly below VEM. We observe a strong correlation between the training instability and the explosion of the value estimation. This result should not come as a surprise since the Adroit tasks have a larger action space compared with MuJoCo domains and narrow human demonstrations. Therefore, the generative model in BCQ cannot guarantee completely the unseen actions are avoided. In contrast, VEM avoids fundamentally unseen actions by keeping the learning procedure within the support of an offline dataset, indicating the necessity of the EVL module. Please refer to Appendix C for the implementation details.

We evaluate $\tau \in \{0.1, 0.2, ..., 0.9\}$ to investigate the effect of the critical hyper-parameter in EVL, which is shown in Figure 7 in Appendix D.1. The experimental results demonstrate that the estimated value increases with a larger $\tau$, which is consistent with the analysis in Section 4.1. Moreover, we observe that $\tau$ is set at a low value in some complex high-dimensional robotic tasks or narrow human demonstrations, such as Adroit-cloned/human, to get the conservative value estimates. However, if $\tau$ is set too high (e.g., $\tau = 0.9$ in the pen-human task), the estimated value will explode and poor performance. This is as expected since the over-large $\tau$ leads to the overestimation error caused by neural networks. The experimental results demonstrate that we can balance behavior cloning and optimal value learning by choosing $\tau$ in terms of different tasks.

### 6.4 ABLATIONS

**Episodic Memory Module.** Our first study aims to answer the impact of memory-based planning on performance. We replace the episodic memory module in VEM with standard $n$-step value estimation (named VEM-1step or VEM-nstep). The experimental results in Figure 8 in Appendix D.1 indicate that implicit planning along offline trajectories effectively accelerates the convergence of EVL.

**Expectile Loss.** In addition to the Expectile loss, we explored other forms of loss. Formally, we compare the Expectile loss and quantile loss, a popular form in Distributional RL algorithms (Dabney et al., 2018), which is shown in Figure 5 in Appendix D.1. The experimental results indicate that the Expectile loss is better since it is more stable when dealing with extreme values.

## 7 CONCLUSION

In this paper, we propose a novel offline RL method, VEM, based on a new $V$-learning algorithm, EVL. EVL naturally avoids actions outside the dataset and provides a smooth tradeoff between generalization and conversation for offline learning. Further, VEM enables effective implicit planning along offline trajectories to accelerate the convergence of EVL and achieve better advantage estimation. Unlike most existing offline RL methods, we keep the learning procedure totally within the dataset's support without any auxiliary modular, such as environment model or behavior policy. The experimental results demonstrate that VEM achieves superior performance in most D4RL tasks and learns the accurate values to guide policy learning, especially in sparse reward tasks. We hope that VEM will inspire more works on offline RL and promote practical RL methods in the future.

## 8 REPRODUCIBILITY

To ensure our work is reproducible, we provide our code in the supplementary materials. In the future, we will publish all source code on Github. The detailed implementation of our algorithm is presented as follows. The value network is trained according to Equation 4. The actor-network is trained according to Equation 7. The hyper-parameters and network structure used in VEM are shown in Appendix C.3. All experiments are run on the standard offline tasks, D4RL (https://github.com/rail-berkeley/d4rl/tree/master/d4rl).

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

# A  ALGORITHM

## A.1  VALUE-BASED EPISODIC MEMORY CONTROL

---
**Algorithm 1** Value-based Episodic Memory Control

---
Initialize critic networks $V_{\theta_1}, V_{\theta_2}$ and actor network $\pi_\phi$ with random parameters $\theta_1, \theta_2, \phi$
Initialize target networks $\theta_1' \leftarrow \theta_1, \theta_2' \leftarrow \theta_2$
Initialize episodic memory $\mathcal{M}$
**for** $t = 1$ **to** $T$ **do**
  **for** $i \in \{1, 2\}$ **do**
    Sample $N$ transitions $\left(s_t, a_t, r_t, s_t, \hat{R}_t^{(i)}\right)$ from $\mathcal{M}$
    Update $\theta_i \leftarrow \min_{\theta_i} N^{-1} \sum \left(R_t^{(i)} - V_{\theta_i}(s_t)\right)^2$
    Update $\phi \leftarrow \max_\phi N^{-1} \sum \nabla \log \pi_\phi(a_t|s_t) \cdot f\left(\min_i \hat{R}_t^{(i)} - \text{mean}_i V_{\theta_i}(s_t)\right)$
  **end for**
  **if** $t \bmod u$ **then**
    $\theta_i' \leftarrow \kappa \theta_i + (1 - \kappa)\theta_i'$
    Update Memory
  **end if**
**end for**

---

---
**Algorithm 2** Update Memory

---
**for** trajectories $\tau$ in buffer $\mathcal{M}$ **do**
  **for** $s_t, a_t, r_t, s_{t+1}$ in reversed($\tau$) **do**
    **for** $i \in \{1, 2\}$ **do**
      Compute $\hat{R}_t^{(i)}$ with Equation 6 and save into buffer $\mathcal{M}$
    **end for**
  **end for**
**end for**

---

## A.2  AN APPROACH FOR AUTO-TUNING $\tau$

When we have a good estimation of $V^*$, for example, when there is some expert data in the dataset, we can auto-tune $\tau$ such that the value learned by EVL is close to the estimation of $V^*$. This can be done by calculating the Monte-Carlo return estimates of each state and selecting good return values as the estimation of optimal value $\tilde{V}^*$. Based on this target, we develop a method for auto-tuning $\tau$.

By parameterizing $\tau = \text{sigmoid}(\xi)$ with a differentiable parameter $\xi \in \mathbb{R}$, we can auto-tune $\tau$ by minimizing the following loss $\mathcal{J}(\xi) = \xi(\mathbb{E}\hat{V}(s) - \tilde{V}^*)$. If $(\mathbb{E}\hat{V}(s) - \tilde{V}^*) < 0$, the differentiable parameter $\xi$ will become larger and the value estimation $\mathbb{E}\hat{V}(s)$ will become larger accordingly. Similarly, $\xi$ and $\mathbb{E}\hat{V}(s)$ will become smaller if $(\mathbb{E}\hat{V}(s) - \tilde{V}^*) > 0$. The experimental results in Figure 10 in Appendix D.1 show that auto-tuning can lead to similar performance compared with manual selection.

# B  THEORETICAL ANALYSIS

## B.1  COMPLETE DERIVATION.

The expectile regression loss (Rowland et al., 2019) is defined as

$$\mathrm{ER}(q; \varrho, \tau) = \mathbb{E}_{Z \sim \varrho}\left[\left[\tau \mathbb{I}(Z > q) + (1-\tau)\mathbb{I}(Z \leq q)\right](Z-q)^2\right], \tag{11}$$

where $\varrho$ is the target distribution and the minimiser of this loss is called the $\tau$-expectile of $\varrho$. the corresponding loss in reinforcement learning is

$$\begin{aligned}
\mathcal{J}_V(\theta) &= \mathbb{E}_\mu\left[\tau(r(s,a) + \gamma V_{\theta'}(s') - V_\theta(s))_+^2 + (1-\tau)(r(s,a) + \gamma V_{\theta'}(s') - V_\theta(s))_-^2\right]\\
&= \mathbb{E}_\mu\left[\tau(y - V_\theta(s))_+^2 + (1-\tau)(y - V_\theta(s))_-^2\right].
\end{aligned} \tag{12}$$

Then, taking the gradient of the value objective with respect to $V_\theta(s)$, we have

$$\begin{aligned}
\nabla \mathcal{J}_V(\theta) &= \sum \mu(a \mid s)\left[-2\tau(y - V_\theta(s))_+ \mathbb{I}(y > V_\theta(s)) - 2(1-\tau)(y - V_\theta(s))_+ \mathbb{I}(y \leq V_\theta(s))\right]\\
&= \sum \mu(a \mid s)\left[-2\tau(y - V_\theta(s))_+ - 2(1-\tau)(y - V_\theta(s))_-\right]\\
&= \sum \mu(a \mid s)\left[-2\tau(\delta)_+ - 2(1-\tau)(\delta)_-\right].
\end{aligned} \tag{13}$$

Therefore,

$$\begin{aligned}
\hat{V}(s) &= V_\theta(s) - \alpha \nabla \mathcal{J}_V(\theta)\\
&= V_\theta(s) + 2\alpha \mathbb{E}_{a \sim \mu}\left[\tau[\delta(s,a)]_+ + (1-\tau)[\delta(s,a)]_-\right]
\end{aligned} \tag{14}$$

## B.2  PROOF OF LEMMA 1

**Lemma 1.** *For any $\tau \in [0,1)$, $\mathcal{T}_\tau^\mu$ is a $\gamma_\tau$-contraction, where $\gamma_\tau = 1 - 2\alpha(1-\gamma)\min\{\tau, 1-\tau\}$.*

*Proof.* Note that $\mathcal{T}_{1/2}^\mu$ is the standard policy evaluation Bellman operator for $\mu$, whose fixed point is $V^\mu$. We see that for any $V_1, V_2$,

$$\begin{aligned}
&\mathcal{T}_{1/2}^\mu V_1(s) - \mathcal{T}_{1/2}^\mu V_2(s)\\
&= V_1(s) + \alpha \mathbb{E}_{a \sim \mu}[\delta_1(s,a)] - (V_2(s) + \alpha \mathbb{E}_{a \sim \mu}[\delta_2(s,a)])\\
&= (1-\alpha)(V_1(s) - V_2(s)) + \alpha \mathbb{E}_{a \sim \mu}[r(s,a) + \gamma V_1(s') - r(s,a) - \gamma V_2(s')]\\
&\leq (1-\alpha)\|V_1 - V_2\|_\infty + \alpha\gamma\|V_1 - V_2\|_\infty\\
&= (1-\alpha(1-\gamma))\|V_1 - V_2\|_\infty.
\end{aligned} \tag{15}$$

We introduce two more operators to simplify the analysis:

$$\begin{aligned}
\mathcal{T}_+^\mu V(s) &= V(s) + \mathbb{E}_{a \sim \mu}[\delta(s,a)]_+,\\
\mathcal{T}_-^\mu V(s) &= V(s) + \mathbb{E}_{a \sim \mu}[\delta(s,a)]_-.
\end{aligned} \tag{16}$$

Next we show that both operators are non-expansion (i.e., $\|\mathcal{T}_+^\mu V_1 - \mathcal{T}_+^\mu V_2\|_\infty \leq \|V_1 - V_2\|_\infty$). For any $V_1, V_2$, we have

$$\begin{aligned}
\mathcal{T}_+^\mu V_1(s) - \mathcal{T}_+^\mu V_2(s) &= V_1(s) - V_2(s) + \mathbb{E}_{a \sim \mu}[[\delta_1(s,a)]_+ - [\delta_2(s,a)]_+]\\
&= \mathbb{E}_{a \sim \mu}[[\delta_1(s,a)]_+ + V_1(s) - ([\delta_2(s,a)]_+ + V_2(s))].
\end{aligned} \tag{17}$$

The relationship between $[\delta_1(s,a)]_+ + V_1(s)$ and $[\delta_2(s,a)]_+ + V_2(s)$ exists in four cases, which are

- $\delta_1 \geq 0, \delta_2 \geq 0$, then $[\delta_1(s,a)]_+ + V_1(s) - ([\delta_2(s,a)]_+ + V_2(s)) = \gamma(V_1(s') - V_2(s'))$.

- $\delta_1 < 0, \delta_2 < 0$, then $[\delta_1(s,a)]_+ + V_1(s) - ([\delta_2(s,a)]_+ + V_2(s)) = V_1(s) - V_2(s)$.

- $\delta_1 \geq 0, \delta_2 < 0$, then

$$\begin{aligned}
&[\delta_1(s,a)]_+ + V_1(s) - ([\delta_2(s,a)]_+ + V_2(s))\\
&= (r(s,a) + \gamma V_1(s')) - V_2(s)\\
&< (r(s,a) + \gamma V_1(s')) - (r(s,a) + \gamma V_2(s'))\\
&= \gamma(V_1(s') - V_2(s')),
\end{aligned} \tag{18}$$

where the inequality comes from $r(s,a) + \gamma V_2(s') < V_2(s)$.

- $\delta_1 < 0, \delta_2 \geq 0$, then

$$
\begin{aligned}
[\delta_1(s,a)]_+ + V_1(s) &- ([\delta_2(s,a)]_+ + V_2(s)) \\
&= V_1(s) - (r(s,a) + \gamma V_2(s')) \\
&\leq V_1(s) - V_2(s),
\end{aligned}
\tag{19}
$$

where the inequality comes from $r(s,a) + \gamma V_2(s') \geq V_2(s)$.

Therefore, we have $\mathcal{T}_+^\mu V_1(s) - \mathcal{T}_+^\mu V_2(s) \leq \|V_1 - V_2\|_\infty$. With the $\mathcal{T}_+^\mu, \mathcal{T}_-^\mu$, we rewrite $\mathcal{T}_\tau^\mu$ as

$$
\begin{aligned}
\mathcal{T}_\tau^\mu V(s) &= V(s) + 2\alpha \mathbb{E}_{a \sim \mu}[\tau[\delta(s,a)]_+ + (1-\tau)[\delta(s,a)]_-] \\
&= (1-2\alpha)V(s) + 2\alpha\tau(V(s) + \mathbb{E}_{a \sim \mu}[\delta(s,a)]_+) + 2\alpha(1-\tau)(V(s) + \mathbb{E}_{a \sim \mu}[\delta(s,a)]_-) \\
&= (1-2\alpha)V(s) + 2\alpha\tau\mathcal{T}_+^\mu V(s) + 2\alpha(1-\tau)\mathcal{T}_-^\mu V(s).
\end{aligned}
\tag{20}
$$

And

$$
\begin{aligned}
\mathcal{T}_{1/2}^\mu V(s) &= V(s) + \alpha \mathbb{E}_{a \sim \mu}[\delta(s,a)] \\
&= V(s) + \alpha(\mathcal{T}_+^\mu V(s) + \mathcal{T}_-^\mu V(s) - 2V(s)) \\
&= (1-2\alpha)V(s) + \alpha(\mathcal{T}_+^\mu V(s) + \mathcal{T}_-^\mu V(s)).
\end{aligned}
\tag{21}
$$

We first focus on $\tau < \frac{1}{2}$. For any $V_1, V_2$, we have

$$
\begin{aligned}
&\mathcal{T}_\tau^\mu V_1(s) - \mathcal{T}_\tau^\mu V_2(s) \\
&= (1-2\alpha)(V_1(s) - V_2(s)) + 2\alpha\tau(\mathcal{T}_+^\mu V_1(s) - \mathcal{T}_+^\mu V_2(s)) + 2\alpha(1-\tau)(\mathcal{T}_-^\mu V_1(s) - \mathcal{T}_-^\mu V_2(s)) \\
&= (1-2\alpha - 2\tau(1-2\alpha))(V_1(s) - V_2(s)) + 2\tau\left(\mathcal{T}_{1/2}^\mu V_1(s) - \mathcal{T}_{1/2}^\mu V_2(s)\right) + \\
&\qquad\qquad\qquad\qquad 2\alpha(1-2\tau)\left(\mathcal{T}_-^\mu V_1(s) - \mathcal{T}_-^\mu V_2(s)\right) \\
&\leq (1-2\alpha - 2\tau(1-2\alpha))\|V_1 - V_2\|_\infty + 2\tau(1 - \alpha(1-\gamma))\|V_1 - V_2\|_\infty + 2\alpha(1-2\tau)\|V_1 - V_2\|_\infty \\
&= (1 - 2\alpha\tau(1-\gamma))\|V_1 - V_2\|_\infty
\end{aligned}
\tag{22}
$$

Similarly, when $\tau > 1/2$, we have $\mathcal{T}_\tau^\mu V_1(s) - \mathcal{T}_\tau^\mu V_2(s) \leq (1 - 2\alpha(1-\tau)(1-\gamma))\|V_1 - V_2\|_\infty$. $\qquad\square$

### B.3 Proof of Lemma 2

**Lemma 2.** *For any $\tau, \tau' \in (0,1)$, if $\tau' \geq \tau$, we have $\mathcal{T}_{\tau'}^\mu \geq \mathcal{T}_\tau^\mu, \forall s \in S$.*

*Proof.* Based on Equation 20, we have

$$
\begin{aligned}
&\mathcal{T}_{\tau'}^\mu V(s) - \mathcal{T}_\tau^\mu V(s) \\
&= (1-2\alpha)V(s) + 2\alpha\tau'\mathcal{T}_+^\mu V(s) + 2\alpha(1-\tau')\mathcal{T}_-^\mu V(s) \\
&\qquad\qquad - ((1-2\alpha)V(s) + 2\alpha\tau\mathcal{T}_+^\mu V(s) + 2\alpha(1-\tau)\mathcal{T}_-^\mu V(s)) \\
&= 2\alpha(\tau' - \tau)(\mathcal{T}_+^\mu V(s) - \mathcal{T}_-^\mu V(s)) \\
&= 2\alpha(\tau' - \tau)\mathbb{E}_{a \sim \mu}[[\delta(s,a)]_+ - [\delta(s,a)]_-] \geq 0.
\end{aligned}
\tag{23}
$$

$\qquad\square$

### B.4 Proof of Lemma 3

**Lemma 3.** *Let $V^*$ denote the fixed point of Bellman optimality operator $\mathcal{T}^*$. In the deterministic MDP, we have $\lim_{\tau \to 1} V_\tau^* = V^*$.*

*Proof.* We first show that $V^*$ is also a fixed point for $\mathcal{T}_+^\mu$. Based on the definition of $\mathcal{T}^*$, we have $V^*(s) = \max_a[r(s,a) + \gamma V^*(s')]$, which infers that $\delta(s,a) \leq 0, \forall s \in S, a \in A$. Thus, we have $\mathcal{T}_+^\mu V^*(s) = V^*(s) + \mathbb{E}_{a \sim \mu}[\delta(s,a)]_+ = V^*(s)$. By setting $(1-\tau) \to 0$, we eliminate the effect of $\mathcal{T}_-^\mu$. Further by the contractive property of $\mathcal{T}_\tau^\mu$, we obtain the uniqueness of $V_\tau^*$. The proof is completed. $\qquad\square$

### B.5 PROOF OF LEMMA 4

**Lemma 4.** *Given $\tau \in (0,1)$ and $T \in \mathbb{N}^+$, $\mathcal{T}_{\text{vem}}$ is a $\gamma_\tau$-contraction. If $\tau > \frac{1}{2}$, $\mathcal{T}_{\text{vem}}$ has the same fixed point as $\mathcal{T}_\tau^\mu$.*

*Proof.* We prove the contraction first. For any $V_1, V_2$, we have

$$
\begin{aligned}
\mathcal{T}_{\text{vem}} V_1(s) - \mathcal{T}_{\text{vem}} V_2(s) &= \max_{1 \leq n \leq n_{\max}} \{(\mathcal{T}^\mu)^{n-1} \mathcal{T}_\tau^\mu V_1(s)\} - \max_{1 \leq n \leq T} \{(\mathcal{T}^\mu)^{n-1} \mathcal{T}_\tau^\mu V_2(s)\} \\
&\leq \max_{1 \leq n \leq n_{\max}} |(\mathcal{T}^\mu)^{n-1} \mathcal{T}_\tau^\mu V_1(s) - (\mathcal{T}^\mu)^{n-1} \mathcal{T}_\tau^\mu V_2(s)| \\
&\leq \max_{1 \leq n \leq n_{\max}} \gamma^{n-1} \gamma_\tau \|V_1 - V_2\|_\infty \\
&\leq \gamma_\tau \|V_1 - V_2\|_\infty.
\end{aligned}
\tag{24}
$$

Next we show that $V_\tau^*$, the fixed point of $\mathcal{T}_\tau^\mu$, is also the fixed point of $\mathcal{T}_{\text{vem}}$ when $\tau > \frac{1}{2}$. By definition, we have $V_\tau^* = \mathcal{T}_\tau^\mu V_\tau^*$. Following Lemma 2, we have $V_\tau^* = \mathcal{T}_\tau^\mu V_\tau^* \geq \mathcal{T}_{1/2}^\mu V_\tau^* = \tilde{\mathcal{T}}^\mu V_\tau^*$. Repeatedly applying $\mathcal{T}^\mu$ and using its monotonicity, we have $\mathcal{T}^\mu V_\tau^* \geq (\mathcal{T}^\mu)^{n-1} V_\tau^*, 1 \leq n \leq n_{\max}$. Thus, we have $\mathcal{T}_{\text{vem}} V_\tau^*(s) = \max_{1 \leq n \leq T} \{(\mathcal{T}^\mu)^{n-1} \mathcal{T}_\tau^\mu V_\tau^*(s)\} = V_\tau^*(s)$. $\qquad \square$

### B.6 PROOF OF LEMMA 5

**Lemma 5.** *When the current value estimates $V(s)$ are much lower than the value of behavior policy, $\mathcal{T}_{\text{vem}}$ provides an optimistic update. Formally, we have*

$$
|\mathcal{T}_{\text{vem}} V(s) - V_\tau^*(s)| \leq \gamma^{n^*(s)-1} \gamma_\tau \|V - V_{n^*,\tau}^\mu\|_\infty + \|V_{n^*,\tau}^\mu - V_\tau^*\|_\infty, \forall s \in S,
\tag{25}
$$

*where $n^*(s) = \arg\max_{1 \leq n \leq T} \{(\mathcal{T}^\mu)^{n-1} \mathcal{T}_\tau^\mu V(s)\}$ and $V_{n^*,\tau}^\mu$ is the fixed point of $(\mathcal{T}^\mu)^{n^*(s)-1} \mathcal{T}_\tau^\mu$.*

*Proof.* The lemma is a direct result of the triangle inequality. We have

$$
\begin{aligned}
\mathcal{T}_{\text{vem}} V(s) - V_\tau^*(s) &= (\mathcal{T}^\mu)^{n^*(s)-1} \mathcal{T}_\tau^\mu V(s) - V_\tau^*(s) \\
&= (\mathcal{T}^\mu)^{n^*(s)-1} \mathcal{T}_\tau^\mu V(s) - (\mathcal{T}^\mu)^{n^*(s)-1} \mathcal{T}_\tau^\mu V_{n^*,\tau}^\mu(s) + V_{n^*,\tau}^\mu(s) - V_\tau^*(s) \\
&\leq \gamma^{n^*(s)-1} \gamma_\tau \|V - V_{n^*,\tau}^\mu\|_\infty + \|V_{n^*,\tau}^\mu - V_\tau^*\|.
\end{aligned}
\tag{26}
$$
$\qquad \square$

### B.7 PROOF OF PROPOSITION 1

**Proposition 1.** *Let $V_\tau^*$ denote the fixed point of $\mathcal{T}_\tau^\mu$. For any $\tau, \tau' \in (0,1)$, if $\tau' \geq \tau$, we have $V_{\tau'}^*(s) \geq V_\tau^*(s), \forall s \in S$.*

*Proof.* With the Lemma 2, we have $\mathcal{T}_{\tau'}^\mu V_\tau^* \geq \mathcal{T}_\tau^\mu V_\tau^*$. Since $V_\tau^*$ is the fixed point of $\mathcal{T}_\tau^\mu$, we have $\mathcal{T}_\tau^\mu V_\tau^* = V_\tau^*$. Putting the results together, we obtain $V_\tau^* = \mathcal{T}_\tau^\mu V_\tau^* \leq \mathcal{T}_{\tau'}^\mu V_\tau^*$. Repeatedly applying $\mathcal{T}_{\tau'}^\mu$ and using its monotonicity, we have $V_\tau^* \leq \mathcal{T}_{\tau'}^\mu V_\tau^* \leq (\mathcal{T}_{\tau'}^\mu)^\infty V_\tau^* = V_{\tau'}^*$. $\qquad \square$

## C DETAILED IMPLEMENTATION

### C.1 GENERALIZED ADVANTAGE-WEIGHTED LEARNING

In practice, we adopt Leaky-ReLU or Softmax functions.

Leaky-ReLU:

$$
\max_\phi J_\pi(\phi) = \mathbb{E}_{(s,a) \sim \mathcal{D}} \left[ \log \pi_\phi(a \mid s) \cdot f\left(\hat{A}(s,a)\right) \right],
$$

$$
\text{where} \quad f(\hat{A}(s,a)) = \begin{cases} \hat{A}(s,a) & \text{if} \quad \hat{A}(s,a) > 0 \\ \frac{\hat{A}(s,a)}{\alpha} & \text{if} \quad \hat{A}(s,a) \leq 0 \end{cases}
\tag{27}
$$

Softmax:

$$\max_{\phi} J_{\pi}(\phi) = \mathbb{E}_{(s,a)\sim\mathcal{D}} \left[ \log \pi_{\phi}(a \mid s) \cdot \frac{\exp(\frac{1}{\alpha}\hat{A}(s,a))}{\sum_{(s_i,a_i)\sim\text{Batch}} \exp(\frac{1}{\alpha}\hat{A}(s_i,a_i))} \right]. \quad (28)$$

## C.2 BCQ-EM

The value network of BCQ-EM is trained by minimizing the following loss:

$$\min_{\theta} \mathcal{J}_Q(\theta) = \mathbb{E}_{(s_t,a_t,s_{t+1})\sim\mathcal{D}} \left[ (R_t - Q_{\theta}(s_t,a_t))^2 \right] \quad (29)$$

$$R_t = \max_{0<n\leq n_{\max}} Q_{t,n}, \quad Q_{t,n} = \begin{cases} r_t + \gamma Q_{t+1,n-1}(s_{t+1},\hat{a}_{t+1}) & \text{if } n > 0, \\ Q(s_t,\hat{a}_t) & \text{if } n = 0, \end{cases} \quad (30)$$

where $\hat{a}_t$ corresponds to the perturbed actions, sampled from the generative model $G_w(s_t)$.

The perturbation network of BCQ-EM is trained by minimizing the following loss:

$$\min_{\phi} \mathcal{J}_{\xi}(\phi) = -\mathbb{E}_{s\sim\mathcal{D}} \left[ Q_{\theta}(s, a_i + \xi_{\phi}(s,a_i,\Phi)) \right], \quad \{a_i \sim G_w(s)\}_{i=1}^{n}, \quad (31)$$

where $\xi_{\phi}(s,a_i,\Phi)$ is a perturbation model, which outputs an adjustment to an action $a$ in the range $[-\Phi,\Phi]$. We adopt conditional variational auto-encoder to represent the generative model $G_w(s)$ and it is trained to match the state-action pairs sampled from $\mathcal{D}$ by minimizing the cross-entropy loss-function.

## C.3 HYPER-PARAMETER AND NETWORK STRUCTURE

Table 2: Hyper-parameter Sheet

| Hyper-Parameter | Value |
|---|---|
| Critic Learning Rate | 1e-3 |
| Actor Learning Rate | 1e-3 |
| Optimizer | Adam |
| Target Update Rate ($\kappa$) | 0.005 |
| Memory Update Period | 100 |
| Batch Size | 128 |
| Discount Factor | 0.99 |
| Gradient Steps per Update | 200 |
| Maximum Length $d$ | Episode Length $T$ |

Table 3: Hyper-Parameter $\tau$ used in VEM across different tasks

| | | | |
|---|---|---|---|
| AntMaze-fixed | umaze | medium | large |
| | 0.4 | 0.3 | 0.3 |
| AntMaze-diverse | umaze | medium | large |
| | 0.3 | 0.4 | 0.1 |
| Adroit-human | door | hammer | pen |
| | 0.4 | 0.4 | 0.4 |
| Adroit-cloned | door | hammer | pen |
| | 0.2 | 0.3 | 0.1 |
| Adroit-expert | door | hammer | pen |
| | 0.3 | 0.3 | 0.3 |
| MuJoCo-medium | walker2d | halfcheetah | hopper |
| | 0.3 | 0.4 | 0.5 |
| MuJoCo-random | walker2d | halfcheetah | hopper |
| | 0.5 | 0.6 | 0.7 |

We use a fully connected neural network as a function approximation with 256 hidden units and ReLU as an activation function. The structure of the actor network is $[(\text{state dim}, 256), (256, 256), (256, \text{action dim})]$. The structure of the value network is $[(\text{state dim}, 256), (256, 256), (256, 1)]$.

# D    ADDITIONAL EXPERIMENTS ON D4RL

## D.1    ABLATION STUDY

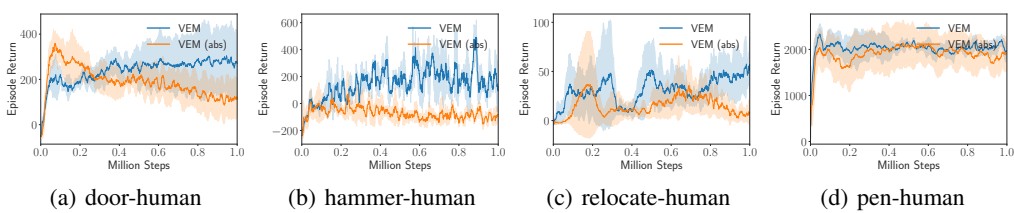

(a) door-human          (b) hammer-human          (c) relocate-human          (d) pen-human

Figure 5: Comparison results between expectile loss and quantile loss on Adroit tasks. We respectively name our algorithm with expectile loss and quantile loss as VEM and VEM (abs).

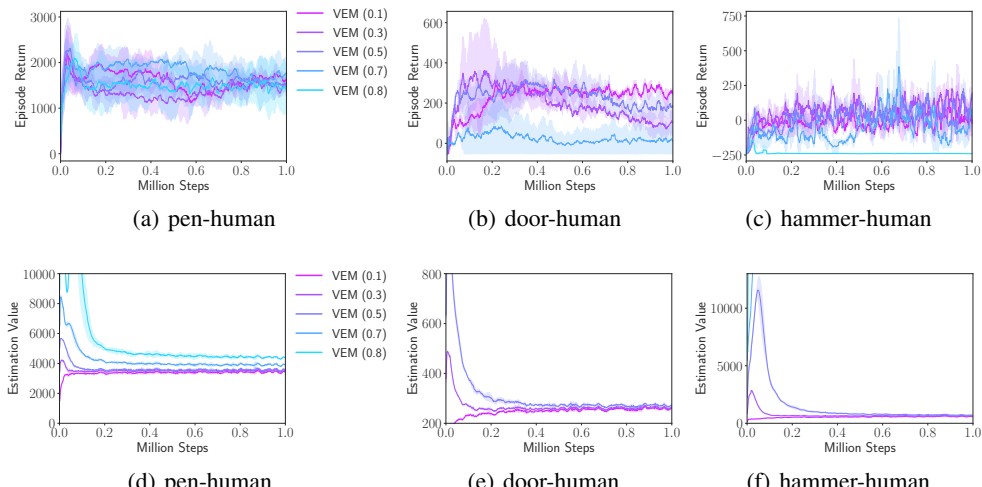

(a) pen-human          (b) door-human          (c) hammer-human

(d) pen-human          (e) door-human          (f) hammer-human

Figure 6: The results of VEM ($\tau$) with various $\tau$ in Adroit tasks. The results in the upper row are the performance. The results in the bottom row are the estimation value.

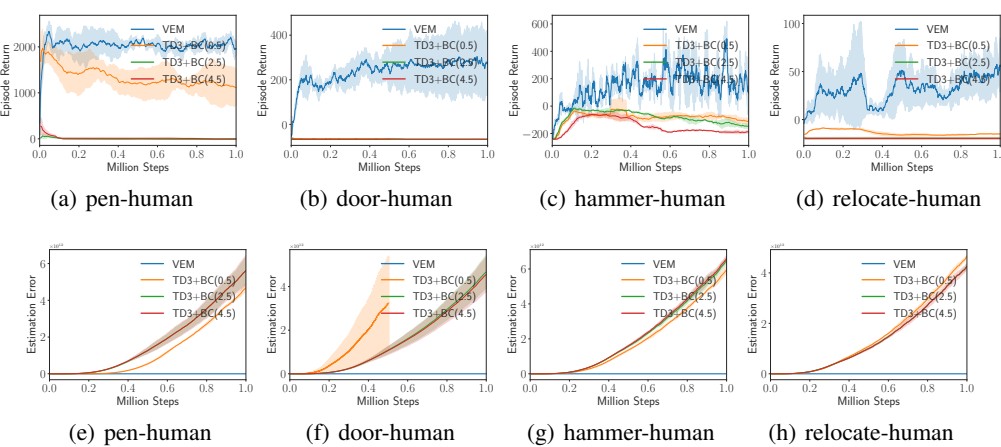

(a) pen-human          (b) door-human          (c) hammer-human          (d) relocate-human

(e) pen-human          (f) door-human          (g) hammer-human          (h) relocate-human

Figure 7: Comparison results between VEM with TD3+BC. We adopt different hyper-parameters $\alpha \in \{0.5, 2.5, 4.5\}$ in TD3+BC to test its performance. The upper row are the performance. The results in the bottom row are the estimation error (the unit is $10^{12}$).

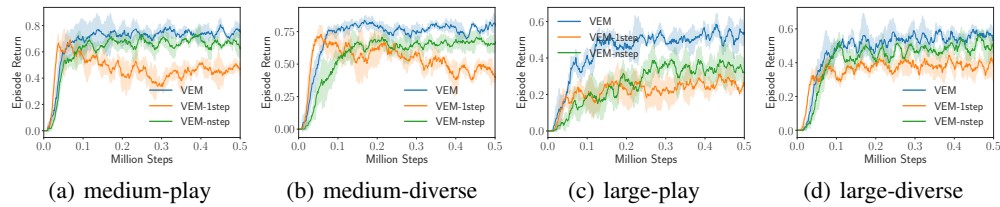

Figure 8: The comparison between episodic memory and $n$-step value estimation on AntMaze tasks.

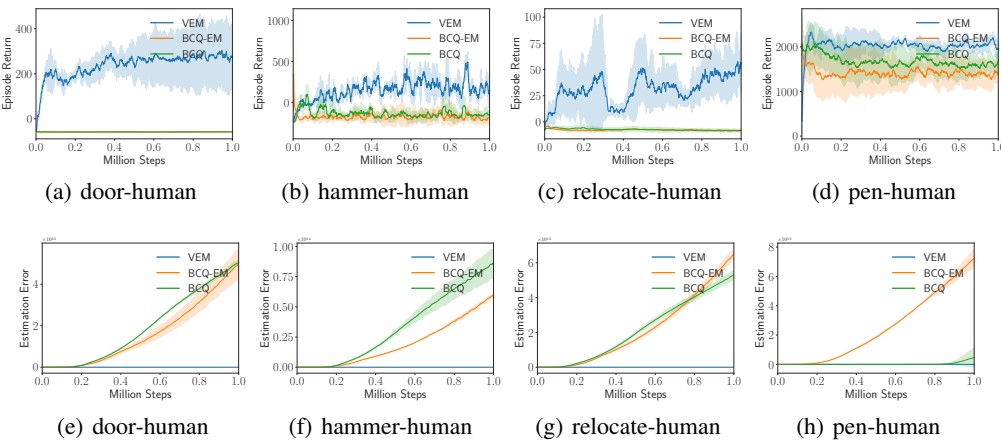

Figure 9: The comparison between VEM, BCQ-EM and BCQ on Adroit-human tasks. The results in the upper row are the performance. The results in the bottom row are the estimation error, where the unit is $10^{13}$.

## D.2 COMPLETE TRAINING CURVES AND VALUE ESTIMATION ERROR

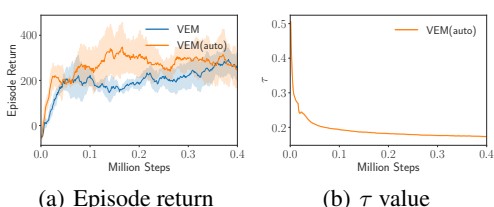

Figure 10: Comparison between fixed $\tau$ (VEM) and auto-tuning $\tau$ (VEM(auto)) in the door-human task.

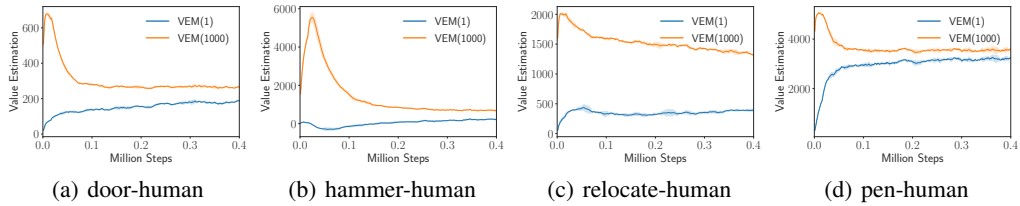

Figure 11: Value estimation of VEM ($n_{\max}$) in adroit-human tasks, where $n_{\max}$ is the maximal rollout step for memory control (see Equation 11). We set $\tau = 0.5$ in all tasks.

(a) door-human   (b) hammer-human   (c) relocate-human   (d) pen-human

(a) antmaze-umaze   (b) antmaze-umaze-diverse   (c) antmaze-medium-play   (d) antmaze-medium-diverse

(e) antmaze-large-play   (f) antmaze-large-diverse   (g) door-human   (h) hammer-human

(i) relocate-human   (j) pen-human   (k) door-cloned   (l) hammer-cloned

(m) pen-cloned   (n) hopper-medium   (o) walker2d-medium   (p) halfcheetah-medium

(q) hopper-random   (r) walker2d-random   (s) halfcheetah-random

Figure 12: The training curves of VEM and BAIL on D4RL tasks.

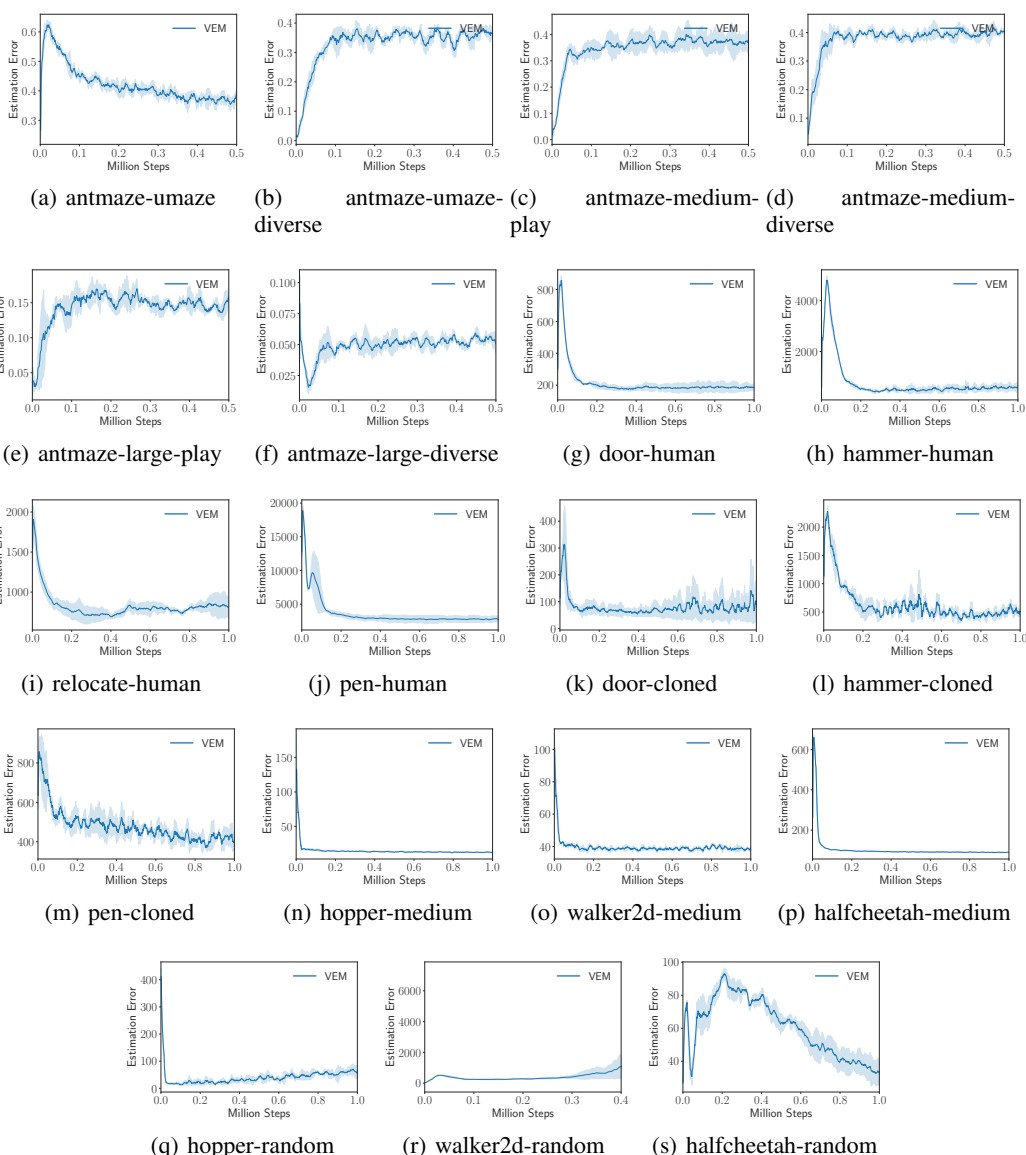

Figure 13: The value estimation error of VEM on D4RL tasks. The estimation error refers to the average estimated state values minus the average returns.

