# OpenReview forum: "Offline Reinforcement Learning with Value-based Episodic Memory"
_ICLR.cc/2022/Conference — ICLR 2022 Poster_

### Official Review · Reviewer_y86L · 2021-11-03

**Correctness:** 3
**Technical Novelty And Significance:** 3
**Empirical Novelty And Significance:** 3
**Recommendation:** 6
**Confidence:** 5

**Main Review:**

Pros:
1. The paper is clearly written and easy to understand.
2. The method is technically sound. The authors provide theoretical guarantees of the convergence of the VEM and also show that the memory-based planning module improves the convergence rate.
3. The empirical results of VEM show that it can outperform prior methods in many of the D4RL tasks.

Cons:
1. I'm not sure why the authors didn't evaluate VEM on the full set of D4RL mujoco datasets, e.g. medium-replay, medium-expert and expert datasets. VEM is also not evaluated on the kitchen dataset. Including these would give a more clear sense of how VEM compares to prior methods.
2. Several important baselines are missing such as [1,2,3,4,5,6]. These methods along with CQL seem to be better than VEM on most of the mujoco tasks while [3] seems to obtain strong adroit results and [6] obtains good antmaze results. The authors should perform a thorough comparison between VEM and these approaches.
3. VEM seems to require per-task tuning with online rollouts as the authors show different $\tau$ values on different tasks/datasets. This could be problematic since it is typically impractical and unsafe for offline RL to evaluate the policy online and per-task tuning would make it even worse.
4. The memory-based planning module seems a bit orthogonal to the main approach, which is the expectile regression. It is directly adapted from prior work and added on top of the method. It would be interesting to see how other offline RL methods perform with the memory-based planning module.

[1] Fujimoto, Scott, and Shixiang Shane Gu. "A Minimalist Approach to Offline Reinforcement Learning." arXiv preprint arXiv:2106.06860 (2021).

[2] Kostrikov, Ilya, et al. "Offline reinforcement learning with fisher divergence critic regularization." International Conference on Machine Learning. PMLR, 2021.

[3] Wu, Yue, et al. "Uncertainty Weighted Actor-Critic for Offline Reinforcement Learning." arXiv preprint arXiv:2105.08140 (2021).

[4] Brandfonbrener, David, et al. "Offline RL Without Off-Policy Evaluation." arXiv preprint arXiv:2106.08909 (2021).

[5] Chen, Lili, et al. "Decision transformer: Reinforcement learning via sequence modeling." arXiv preprint arXiv:2106.01345 (2021).

[6] Ajay, Anurag, et al. "Opal: Offline primitive discovery for accelerating offline reinforcement learning." arXiv preprint arXiv:2010.13611 (2020).

**Summary Of The Paper:**

This paper presents a new offline RL algorithm that leverages expectile regression for value learning and performs AWR-style policy learning with the value function learned with the expectile loss and memory-based planning. The proposed method, dubbed VEM, is able to interpolate between learning optimal Bellman operators and behavior cloning, preventing overestimation in Bellman backups. VEM is shown to converge and the episodic memory-based planning module is able to both theoretically improve the convergence rate and also empirically improve the performance. VEM achieves the best or comparable performance on most of the D4RL tasks.

**Summary Of The Review:**

Given the comments in the previous section, I think it would be good if the authors can address the raised issues. Based on its current status, I would vote for a weak reject.

_____________________________
post rebuttal update: I've read the response and most of my concerns are addressed. I will raise my score to a 6.

---

> ### Author Response · Authors · 2021-11-21
> **Response to Reviewer y86L (Part I)**
>
> Dear Reviewer:
>
> Thanks for your detailed and valuable comments.
> We add supplementary experimental results on D4RL tasks as part of our response and hope our clarification addresses your concerns.
>
> **Q1: Lack of experiments on the full set of D4RL mujoco datasets and kitchen dataset.**
>
> **A1:**
> As suggested by the reviewer, we conducted additional experiments in mujoco and kitchen tasks, and the results are shown in the following table.
> Combining these results with initial results in other settings in the paper, we can see that our method VEM significantly outperforms baselines in more challenging tasks such as antmaze and adroit tasks, and achieves better or comparable performance in the kitchen dataset and most mujoco tasks.
>
> | **Dataset Type** | **Environments** | **VEM(Ours)**     | BAIL                  | BCQ   | CQL            | AWR  |
> |---------------------|-----------------------|------------------------|-----------------------|-------|----------------|------|
> | random              | mujoco-walker2d       | 6.2$\pm$4.7            | 3.9$\pm$2.5           | 4.9   | **7.0**   | 1.5  |
> | random              | mujoco-hopper         | **11.1$\pm$1.0**  | 9.8$\pm$0.1           | 10.6  | 10.8           | 10.2 |
> | random              | mujoco-halfcheetah    | 16.4$\pm$3.6           | 0.0$\pm$0.1           | 2.2   | **35.4**  | 2.5  |
> | medium              | mujoco-walker2d       | 74.0$\pm$1.2           | 73.0$\pm$1.0          | 53.1  | **79.2**  | 17.4 |
> | medium              | mujoco-hopper         | 56.6$\pm$2.3           | **58.2$\pm$1.0** | 54.5  | 58.0           | 35.9 |
> | medium              | mujoco-halfcheetah    | **47.4$\pm$0.2**  | 42.6$\pm$1.2          | 40.7  | 44.4           | 37.4 |
> | medium-replay       | mujoco-walker2d       | **56.6$\pm$4.2**  | 46.6$\pm$4.6          | 15.0  | 26.7           | 15.5 |
> | medium-replay       | mujoco-hopper         | **74.2**$\pm$6.8  | 48.7$\pm$1.9          | 33.1  | 48.6           | 28.4 |
> | medium-replay       | mujoco-halfcheetah    | 38.7$\pm$2.3           | 38.6$\pm$2.5          | 38.2  | **46.2**  | 40.3 |
> | medium-expert       | mujoco-walker2d       | **110.5$\pm$5.7** | 109.4$\pm$2.3         | 57.5  | 98.7           | 53.8 |
> | medium-expert       | mujoco-hopper         | 95.8$\pm$3.7           | 29.4$\pm$6.6          | 110.9 | **111.0** | 27.1 |
> | medium-expert       | mujoco-halfcheetah    | **87.3$\pm$3.4**  | 81.2$\pm$3.5          | 64.7  | 62.4           | 52.7 |
> | expert              | mujoco-walker2d       | 111.0$\pm$2.1          | 75.2$\pm$4.1          | -     | **153.9** |      |
> | expert              | mujoco-hopper         | 105.2$\pm$3.5          | 98.2$\pm$0.6          | -     | **109.9** | -    |
> | expert              | mujoco-halfcheetah    | **109.6$\pm$1.3** | 80.9$\pm$1.3          | -     | 104.8          | -    |
> | kitchen             | complete              | **68.1$\pm$5.8**  | 52.2$\pm$3.7           | 8.1   | 43.8           | 0.0  |
> | kitchen             | partial               | 45.5$\pm$3.9           | 45.0$\pm$4.4          | 18.9  | **50.1**  | 15.4 |
> | kitchen             | mixed                 | **54.5$\pm$4.2**  | 50.1$\pm$5.7          | 8.1   | 52.4           | 10.6 |
>
> Table 1. Comparison between VEM and four offline RL methods in mujoco and kitchen tasks. The results of BCQ and AWR are taken from [1].
> The results of CQL are taken from [5].
> We reproduce BAIL according to the author's code: https://github.com/lanyavik/BAIL.

---

> > ### Author Response · Authors · 2021-11-21
> > **Response to Reviewer y86L (Part II)**
> >
> > **Q2: Lack of thorough comparison with several important baselines.**
> >
> > **A2:**
> > As suggested by the reviewer, we conduct a thorough comparison between VEM with other baselines in complex tasks, shown in the following table.
> > VEM achieves state-of-the-art performance in most adroit tasks, although UWAC [3] obtains strong adroit results.
> > Furthermore, VEM achieves superior performance in most antmaze tasks without any auxiliary model, prior learning, or constraints, making it a simple yet powerful method against existing baselines.
> > However, OPAL [4] needs to use prior knowledge to obtain good performance in antmaze tasks, limiting their applications to more general scenarios.
> >
> > | **Type** | **Environments** | **VEM(Ours)**     | TD3+BC       | UWAC           | OAMPI                 |
> > |-------------|-----------------------|------------------------|--------------|----------------|-----------------------|
> > | fixed       | antmaze-umaze         | **87.5$\pm$1.1**  | 71.0$\pm$5.0 | -  | 82.0$\pm$2.7          |
> > | play        | antmaze-medium        | **78.0$\pm$3.0**  | 51.0$\pm$2.5 |  -  | 75.0$\pm$1.3          |
> > | play        | antmaze-large         | **57.0$\pm$5.0**  | 10.0$\pm$3.0 |  -  | 32.0$\pm$4.6          |
> > | diverse     | antmaze-umaze         | 78.0$\pm$1.1           | 68.5$\pm$3.5 |  -   | **85.0$\pm$4.2** |
> > | diverse     | antmaze-medium        | **77.0$\pm$2.2**  | 61.0$\pm$2.5 |   -   | 70.5$\pm$3.9          |
> > | diverse     | antmaze-large         | **58.0$\pm$2.1**  | 3.0$\pm$0.5  |   -    | 25.0$\pm$5.2          |
> > | human       | adroit-door           | **11.2$\pm$4.2**  | 0.0$\pm$0.0  | 10.7           | 2.8$\pm$0.8           |
> > | human       | adroit-hammer         | 3.6$\pm$1.0            | 0.9$\pm$0.2  | **8.3**   | 3.9$\pm$3.0           |
> > | human       | adroit-relocate       | **1.3$\pm$0.2**   | 0.0$\pm$0.0  | 0.5            | 0.5$\pm$0.2           |
> > | human       | adroit-pen            | **65.0$\pm$2.1**  | 0.0$\pm$0.0  | **65.0**  | 54.6$\pm$18.7         |
> > | cloned      | adroit-door           | **3.6$\pm$0.3**   | 0.0$\pm$0.0  | 1.2            | 0.0$\pm$0.0           |
> > | cloned      | adroit-hammer         | **2.7$\pm$1.5**   | 0.0$\pm$0.0  | 1.2            | 0.1$\pm$0.1           |
> > | cloned      | adroit-pen            | **48.7$\pm$3.2**  | 0.0$\pm$0.1  | 45.1           | 23.9$\pm$3.4          |
> > | expert      | adroit-door           | **105.5$\pm$0.2** | 0.0$\pm$0.2  | **105.4** | 102.4$\pm$2.7         |
> > | expert      | adroit-hammer         | **128.3$\pm$1.1** | 3.1$\pm$1.5  | **128.8** | 0.8$\pm$0.2           |
> > | expert      | adroit-relocate       | **109.8$\pm$0.2** | 0.0$\pm$0.0  | 108.7          | 0.6$\pm$0.3           |
> > | expert      | adroit-pen            | 111.7$\pm$2.6          | 52.0$\pm$3.7 | **119.8** | 5.1$\pm$1.8           |
> >
> > Table 2. Comparison between VEM and four important offline RL methods in adroit and antmaze tasks. The results of UWAC are taken from [3]. We reproduce TD3+BC according to the author's code: https://github.com/sfujim/TD3_BC. We reproduce OAMPI according to the author's code: https://github.com/davidbrandfonbrener/onestep-rl.
> > Due to the limited time, we will add the performance of UWAC in antmaze tasks soon.

---

> > > ### Author Response · Authors · 2021-11-21
> > > **Response to Reviewer y86L (Part III)**
> > >
> > > **Q3: How to choose $\tau$ in practice.**
> > >
> > > **A3:**
> > > This is an important question, and we thank the reviewer for pointing it out.
> > > Here we give two guidelines for choosing $\tau$ without online interactions.
> > >
> > > - **The principle for selecting $\tau$.**
> > > As depicted in Section 3.1, the choice of $\tau$ is mainly affected by the amount of noise in the expectile operator (defined by Equation 2) when applied to an offline dataset.
> > > The small amount of data, long horizon, and sparse reward environment can lead to high variance for samples in the dataset.
> > > So we recommend smaller $\tau$ in these scenarios to keep a conservative estimation and larger $\tau$ when you have large datasets, short-horizon tasks, or dense reward settings.
> > > Furthermore, we empirically find that the choice of $\tau$ is quite robust, i.e., a small change of the $\tau$ value has little effect on the performance, and thus we do not need to adjust $\tau$ very precisely.
> > >
> > > - **An approach for auto-tuning $\tau$.**
> > > When we have a good estimation of $V\^* $ (e.g., by utilizing some trajectories with high returns in the offline dataset), we can auto-tune $\tau$ such that the value learned by EVL is close to the estimation of $V\^* $. This can be done by calculating the Monte-Carlo return estimates of each state and selecting good return values as the estimation of optimal value $\tilde{V}^*$.
> > > Based on this target, we develop a method for auto-tuning $\tau$, shown in Appendix A.2.
> > > The experimental results in Figure 9 in Appendix D.1 show that auto-tuning can lead to similar performance compared with manual selection.
> > >
> > > **Q4: The memory-based planning module seems orthogonal to the expectile regression.
> > > How other offline RL methods perform with the memory-based planning module?**
> > >
> > > **A4:**
> > > They are not purely orthogonal. In order to perform effectively in offline settings, it is crucial for memory-based planning to utilize the expectile value learning (EVL) since the original memory-based planning will magnify the extrapolation error due to its overestimation nature. We compared our method with BCQ+EM in Figure 9 in Appendix D.1. The results show that BCQ+EM has a significant estimation error and leads to suboptimal performance. Instead, VEM successfully confines planning within the dataset and reduces overestimation by controlling the trade-off between conservatism and generalization. We also provide a theoretical analysis of the combination of EVL and EM in Lemma 4 and Lemma 5, revealing that EM can accelerate the convergence without additional bias on EVL.
> > >
> > > Thanks again for your supportive comments and suggestions.
> > > We sincerely hope that our response has cleared your concerns regarding the experiments and algorithm.
> > > We are looking forward to more discussions.
> > >
> > > [1] Fu, Justin, et al. "D4rl: Datasets for deep data-driven reinforcement learning." arXiv preprint arXiv:2004.07219 (2020).
> > >
> > > [2] Fujimoto, Scott, and Shixiang Shane Gu. "A Minimalist Approach to Offline Reinforcement Learning." arXiv preprint arXiv:2106.06860 (2021).
> > >
> > > [3] Wu, Yue, et al. "Uncertainty Weighted Actor-Critic for Offline Reinforcement Learning." arXiv preprint arXiv:2105.08140 (2021).
> > >
> > > [4] Ajay, Anurag, et al. "Opal: Offline primitive discovery for accelerating offline reinforcement learning." arXiv preprint arXiv:2010.13611 (2020).
> > >
> > > [5] Kumar, Aviral, et al. "Conservative q-learning for offline reinforcement learning." arXiv preprint arXiv:2006.04779 (2020).
> > >
> > > [6] Brandfonbrener, David, et al. "Offline RL Without Off-Policy Evaluation." arXiv preprint arXiv:2106.08909 (2021).
> > >
> > > [7] Chen, Lili, et al. "Decision transformer: Reinforcement learning via sequence modeling." arXiv preprint arXiv:2106.01345 (2021).

---

> ### Author Response · Authors · 2021-11-26
> **Looking forward to further discussions!**
>
> Dear reviewer,
>
> We were wondering if our response and revision have resolved your concerns.
> In our responses, we gave a thorough comparison between our method and several important baselines on the full set of D4RL mujoco/adroit/antmaze/kitchen datasets. Please let us know if you have any further concerns, and we are looking forward to more discussions to improve our manuscript further.
>
> Best regards,
>
> The Authors

---

> ### Author Response · Authors · 2021-11-28
> **Thank you for your time and efforts in reviewing our work !**
>
> Dear Reviewer,
>
> Thank you for your time and efforts in reviewing our work. We have provided detailed clarification and additional experiments to address the issues raised in your comments. If our response has addressed your concerns, we would be grateful if you could re-evaluate our work.
>
> If you have any additional questions or comments, we would be happy to have further discussions.
>
> Thanks,
>
> The authors

---

> ### Author Response · Authors · 2021-12-03
> **Thanks for raising the score!**
>
> We would like to thank the reviewer for raising the score!
> We really appreciate the valuable comments and suggestions from the reviewer.

---

### Official Review · Reviewer_3Tyw · 2021-11-06

**Correctness:** 4
**Technical Novelty And Significance:** 3
**Empirical Novelty And Significance:** 2
**Recommendation:** 8
**Confidence:** 4

**Main Review:**

Strengths:
1. The paper proposes a principled way to prevent overestimation in value function in offline RL by using expectile value function learning.
2. It proposes using episodic memory to obtain better estimates of returns for AWR during policy learning phase
3. The paper provides theoritical guarantees for the method and shows improved performance on subset of D4RL tasks.

Concerns:
The proposed method feels like implicit/non-parameteric offline model based RL given it uses implicit planning to obtain better targets during policy learning phase. Hence, my main concern is that authors should compare their proposed method to model based offline RL methods like MoREL (kidambi et al., 2020), Combo (Yu et al., 2021).

References:
1. MOReL: Model-Based Offline Reinforcement Learning. Kidami et al., 2020.
2. COMBO: Conservative Offline Model-Based Policy Optimization. Yu et al., 2021.

**Summary Of The Paper:**

The paper proposes to reduce the overestimation in offline reinforcement learning by (i) learning $V(s)$ function instead of $Q(s,a)$ (and then using AWR to get the policy) to be within support of data and (ii) using expectile learning to train value function that interpolates between optimal value learning and BC. Furthermore, to obtain better advantage estimation for AWR during policy learning, the paper proposes to compare best return along the trajectory with value estimates and take the maximum between the two (i.e. $R_t = r_t + \max(R_{t+1}, V(s_{t+1}))$). Finally, the paper provides theoretical guarantees for the proposed method and shows improved performance on a subset of D4RL tasks.

**Summary Of The Review:**

Weighing the strengths and concerns of the paper, I am recommending weak accept as of now.

---

> ### Author Response · Authors · 2021-11-21
> **Response to Reviewer 3Tyw**
>
> Dear Reviewer:
>
> Thanks for your detailed and valuable comments.
> We provide clarification to your questions and concerns as below. If you have any further questions or comments, we will be happy to have further discussions.
>
> **Q1: Comparison of VEM with model based offline RL methods like MoREL and COMBO.**
>
> **A1:**
> As suggested by the reviewer, we conducted this comparison, and the experimental results are shown in the following table.
> VEM significantly outperforms MoREL and COMBO in more challenging tasks such as antmaze and adroit tasks and achieves better or comparable performance in most mujoco tasks.
> Note that adroit tasks with high-dimension action space are relatively difficult.
> Antmaze tasks have sparse rewards and long horizons.
> The datasets of mujoco tasks are large, and the action space is relatively low-dimension.
>
> Although the memory-based planning in VEM has some similarities with implicit/non-parametric offline model-based RL, there are some fundamental differences between them:
>
> - VEM adopts the EVL operator to achieve the trade-off between imitation learning and optimal value learning without any auxiliary models or constraints.
> This measure avoids using actions out of the dataset and significantly alleviates the extrapolation error.
> Furthermore, the memory-based planning module in VEM helps obtain better target estimates for EVL.
> Therefore, VEM achieves significantly better performance in complex tasks, such as high-dimensional manipulation tasks (adroit) and sparse rewards tasks (antmaze).
>
> - On the contrary, model-based offline RL methods learn a pessimistic MDP using the offline dataset and perform the pessimistic planning under the learned MDP.
> However, the estimation error introduced by the auxiliary model in high-dimensional manipulation tasks can be enormous, which leads to poor performance in adroit tasks.
> Differently, mujoco tasks have small action space and provide enough data to train the MDP model.
> Therefore, MoReL and COMBO achieve good performance in mujoco tasks.
>
> | Type | Environments | VEM(Ours) | MoReL | COMBO |
> | ---- | :---: | :---: | :---: | :---: |
> | fixed | antmaze-umaze | **87.5$\pm$1.1** | 0.0 | 80.3 |
> | play | antmaze-medium | **78.0$\pm$3.0** | 0.0 | 0.0 |
> | play | antmaze-large | **57.0$\pm$5.0** | 0.0 | 0.0 |
> | diverse | antmaze-umaze | **78.0$\pm$1.1** | 0.0 | 57.3 |
> | diverse | antmaze-medium | **77.0$\pm$2.2** | 0.0 | 0.0 |
> | diverse | antmaze-large | **58.0$\pm$2.1** | 0.0 | 0.0 |
> | human | adroit-door | **11.2$\pm$4.2** | 0.0$\pm$0.0 | 0.0$\pm$0.0 |
> | human | adroit-hammer | 3.6$\pm$1.0 | 0.0$\pm$0.0 | 0.0$\pm$0.0 |
> | human | adroit-relocate | **1.3$\pm$0.2** | 0.0$\pm$0.0 | 0.0$\pm$0.0 |
> | human | adroit-pen | **65.0$\pm$2.1** | 0.0$\pm$0 | 8.5$\pm$0.1 |
> | cloned | adroit-door | **3.6$\pm$0.3** | 0.0$\pm$0.0 | 0.0$\pm$0.0 |
> | cloned | adroit-hammer | **2.7$\pm$1.5** | 0.0$\pm$0.0 | 0.0$\pm$0.0 |
> | cloned | adroit-pen | **48.7$\pm$3.2** | 1.6$\pm$1.1 | 0.0$\pm$0.0 |
> | expert | adroit-door | **105.5$\pm$0.2** | 0.0$\pm$0.2 | 4.9$\pm$1.2 |
> | expert | adroit-hammer | **128.3$\pm$1.1** | 0.0$\pm$0.1 | 0.2$\pm$0.1 |
> | expert | adroit-relocate | **109.8$\pm$0.2** | 0.0$\pm$0.1 | 17.2$\pm$3.1 |
> | expert | adroit-pen | **111.7$\pm$2.6** | 0.0$\pm$0.1 | 11.5$\pm$2.3 |
> | random | mujoco-walker2d | 6.2$\pm$4.7 | **37.3** | 7.0 |
> | random | mujoco-hopper | 11.1$\pm$1.0 | **53.6** | 17.9 |
> | random | mujoco-halfcheetah | 16.4$\pm$3.6 | 25.6 | **38.8** |
> | medium | mujoco-walker2d | 74.0$\pm$1.2 | **77.8** | 75.5 |
> | medium | mujoco-hopper | 56.6$\pm$2.3 | **95.4** | 94.9 |
> | medium | mujoco-halfcheetah | 47.4$\pm$0.2 | 42.1 | **54.2** |
> | medium-replay | mujoco-walker2d | **56.6$\pm$4.2** | 49.8 | 56.0 |
> | medium-replay | mujoco-hopper | 74.2$\pm$6.8 | **93.6** | 73.1 |
> | medium-replay | mujoco-halfcheetah | 38.7$\pm$2.3 | 40.2 | **55.1** |
> | medium-expert | mujoco-walker2d | **110.5$\pm$5.7** | 95.6 | 96.1 |
> | medium-expert | mujoco-hopper | 95.8$\pm$3.7 | 108.7 | **111.1** |
> | medium-expert | mujoco-halfcheetah | 87.3$\pm$3.4 | 53.3 | **90.0** |
>
> Table 1. Comparison between VEM and model-based offline RL methods (MOReL and COMBO).
> The experimental results of MOReL and COMBO in mujoco tasks are respectively taken from [1] and [2].
> We reproduce MOReL according to the author's code: https://github.com/SwapnilPande/MOReL.
> We reproduce COMBO according to an offline learning library: https://github.com/takuseno/d3rlpy.
> The experimental results of MOReL and COMBO in antmaze tasks are taken from [3].
>
> Thanks again for the valuable comments.
> We sincerely hope our response has cleared your remaining concerns.
>
> [1] MOReL: Model-Based Offline Reinforcement Learning. Kidami et al., 2020.
>
> [2] COMBO: Conservative Offline Model-Based Policy Optimization. Yu et al., 2021.
>
> [3] Wang, Jianhao, et al. "Offline Reinforcement Learning with Reverse Model-based Imagination." Advances in Neural Information Processing Systems 34 (2021).

---

> ### Author Response · Authors · 2021-11-26
> **Looking forward to further discussions!**
>
> Dear reviewer,
>
> We were wondering if our response and revision have resolved your concerns.
> In our responses, we gave a thorough comparison with two model-based offline methods and made our contribution more clear. Please let us know if you have any further concerns, and we are looking forward to more discussions to improve our manuscript further.
>
> Best regards,
>
> The Authors

---

> ### Author Response · Authors · 2021-11-28
> **Thank you for your time and efforts in reviewing our work !**
>
> Dear Reviewer,
>
> Thank you for your time and efforts in reviewing our work. We have provided detailed clarification and additional experiments to address the issues raised in your comments. If our response has addressed your concerns, we would be grateful if you could re-evaluate our work.
>
> If you have any additional questions or comments, we would be happy to have further discussions.
>
> Thanks,
>
> The authors

---

> ### Comment · Reviewer_3Tyw · 2021-11-29
> **Thanks for resolving my concerns**
>
> I would like to thank the authors for resolving all my concerns. I would like to raise the score to 8.

---

> ### Author Response · Authors · 2021-11-29
> **Thanks for raising the score to 8!**
>
> We would like to thank the reviewer for raising the score!
> We really appreciate the valuable comments and suggestions from the reviewer.

---

### Official Review · Reviewer_M22A · 2021-11-07

**Correctness:** 3
**Technical Novelty And Significance:** 2
**Empirical Novelty And Significance:** 3
**Recommendation:** 6
**Confidence:** 3

**Main Review:**

Advantages:
1. This paper provides a new perspective to evaluate actions out of the dataset's support. Traditional attempts focused on Q-based methods which require additional constraints or penalties for actions out of the dataset. This paper learns bootstrapped V-values while being completely confined within the dataset without any regularization.
2. The Value-based Episodic Memory is simple yet efficient. The value-based planning to conduct bootstrapping is efficient. The adaption of episodic memory-based methods is appropriate. The adaption of return-based offline RL methods is effective.
3. The theoretical analysis is convincing and enhanced the persuasion of the claims.
4. The experiments are extensive and supportive. The critical experimental parameters are provided, and thus there should be no issues with the repeatability of the experiments.

Disadvantages:
1. There are quite a few contributions of this paper, but none of them are significant. The first contribution (the most important one), using V-function to substitute Q-function, doesn't show distinct advantages intuitively. Why the V-function is better than Q-function? Simply by adding no additional constraint or penalty for actions out of the dataset? If so, what are the basic reasons? Is it because the constraint or penalty in Q-function difficult to tackle or not reasonable? The contribution of balance between imitation learning and optimal value learning is good, but it is also trivial by just combining two losses together. The implicit memory-based planning and generalized advantage-weighted learning are direct adaptions from existing work.
2. The Equation formations should be unified. For example, Equation (2) should include (s,a) in the Dirac function.

**Summary Of The Paper:**

In this paper, the authors intend to derive offline reinforcement learning policies by learning the V-function, instead of the Q-function, so as to balance the imitation learning and optimal value learning. To achieve this goal, the author proposes Expectile V-learning (EVL) to smoothly interpolate between the Bellman expectation operator and optimality operator. Based on the learned value, the authors propose Value-based Episodic Memory (VEM) to approximate the optimal value with offline data and conduct implicit memory-based planning to further enhance advantage estimation. The authors design theoretical analysis, a toy example, and empirical experiments to validate the proposed methods.

**Summary Of The Review:**

The Expectile V-learning (EVL) is new, the Value-based Episodic Memory is simple yet efficient, the theoretical analysis is convincing, and the experiments are supportive. However, my major concern is that none of the contributions are significant. I am looking forward to the authors' response, especially justifications on the novelty and effectiveness of  EVL.

---

> ### Author Response · Authors · 2021-11-21
> **Response to Reviewer M22A (Part I)**
>
> Dear Reviewer:
>
> Thank you for your insightful comments. We conduct additional experiments to clarify your questions and concerns. We sincerely hope you can re-evaluate our work based on the updated information. If you have any additional questions, we will be happy to have further discussions.
>
> **Q1: The superiority of $V$-learning over $Q$-learning that requires penalty or constraint for unseen actions.**
>
> **A1:**
> The superiority of $V$-learning comes from the ability to confine learning within the dataset, which induces a stronger conservationism bias, while $Q$-based methods that confine learning within the distribution of the dataset. Empirically, we observe that in-distribution constraint can be insufficient to deal with over-generalization of the value function extrapolation.
> To illustrate this, we compare BCQ and EVL in a random MDP.
> We randomly collect half of the state-action pairs to create an offline dataset. We assume a small error when estimating behavior policy with a finite dataset,  i.e., $\hat{\mu} = (1 - \epsilon) \mu + \epsilon \mu\_\epsilon$, where $\mu$ is the actual behavior and $\mu\_\epsilon$ is a random policy. We confine policy within a small range $\zeta$ from estimated $\hat{\mu}$, and we compare BCQ with different constraint coefficients $\zeta$ and EVL. The results are shown as follows. Note that $\zeta=0$ is infeasible in continuous settings.
>
> | $\zeta$ | $  \parallel Q - Q\^* \parallel\_\infty $ |
> | ------- | ---------------------- |
> | 0\% | $0.27 \pm 0.06$ |
> | 2\% | $1.89 \pm 1.13$ |
> | 4\% | $2.67 \pm 1.08$ |
> | 6\% | $2.81 \pm 1.09$ |
> | 8\% | $3.05 \pm 0.93$ |
> | 10\% | $3.37 \pm 0.95$ |
> | 20\% | $3.81 \pm 0.85$ |
> | 40\% | $4.17 \pm 0.72$ |
>
> Table 1. Toy example results of BCQ. (Averaged over 100 seeds)
>
> | $\tau $ | $ \parallel V - V\^* \parallel\_\infty $ |
> | ------- | ----------------------- |
> | 0.5 | $3.61 \pm 0.24$ |
> | 0.6 | $2.84 \pm 0.22$ |
> | 0.7 | $2.10 \pm 0.22$ |
> | 0.8 | $1.29 \pm 0.24$ |
> | 0.9 | $0.40 \pm 0.15$ |
> | 0.95 | $1.07 \pm 0.18$ |
> | 0.98 | $2.02 \pm 0.18$ |
>
> Table 2. Toy example results of EVL. (Averaged over 100 seeds)
>
> We can see that as $\zeta$ increases, the value estimation also increases in a monotonical manner. Instead, EVL achieves the smallest error with proper $\tau$. This is aligned with our empirical findings. As shown in Figure 8 in Appendix D.1 in the updated paper, BCQ has significant value errors in adroit tasks with a high-dimension action space and small data. In these kinds of tasks, extrapolation in action space can lead to unrealistic value estimates. This phenomenon does not only exist in BCQ but a wide range of $Q$-based methods. We also evaluate TD3+BC in adroit (shown in Figure 6), and we observe a similar phenomenon in its value error.
>
> **Q2: The novelty and effectiveness of EVL. Combining imitation learning and optimal value learning losses is trivial.**
>
> **A2:**
> We respectfully disagree with this statement. Effectively combining imitation learning and optimal value learning losses is non-trivial, and the trade-off achieved by EVL is novel and significant, as discussed for the following reasons:
>
> - The novelty of EVL is that it achieves the trade-off between imitation learning and optimal value learning without using the maximization operation over the actions.
> In contrast, if we maximize values over actions, the unseen actions will inevitably be introduced, especially in the continuous action space.
> As stated in A1, the involved unseen actions can have over-estimated values and lead to sub-optimal performances.
>
> - Some works directly combine imitation learning and optimal value learning losses, such as TD3+BC[1], which achieves sound performance in mujoco tasks.
> However, the estimated value of TD3+BC quickly explodes in complex tasks such as adroit-human (Please refer to the Figure 6 in Appendix D.1 in the updated paper).
> Instead, EVL has good results across these complex tasks.
> It learns value function without prior knowledge, constraints, and uncertainty estimation, making it a simple yet powerful method against existing baselines.

---

> > ### Author Response · Authors · 2021-11-21
> > **Response to Reviewer M22A (Part II)**
> >
> > **Q3: The role of implicit memory-based planning and generalized advantage-weighted learning.**
> >
> > **A3:**
> > As pointed out by the reviewer, our methods belong to the category of advantage-weighted learning-based methods. However, our method novelly introduces expectile value learning (EVL) for better value estimation and effectively adapts memory-based planning to boost learning in offline settings.
> >
> > Directly adopting them in offline RL is unavailable since memory-based planning will magnify the extrapolation error due to its overestimation nature. We compared our method with BCQ+EM in Figure 8 in Appendix D.1. The result shows that BCQ+EM has a significant estimation error and leads to suboptimal performance. Instead, VEM successfully confines planning within the dataset and reduces overestimation by controlling the trade-off between conservatism and generalization. We also provide a theoretical analysis of the combination of EVL and EM in Lemma 4 and Lemma 5, revealing that EM can accelerate the convergence without additional bias on EVL.
> >
> > **Q4: Unifying formation of equations.**
> >
> > **A4:**
> > Thanks for your detailed comments. We unified the formations in the updated paper according to your suggestions.
> >
> > Thanks again for the valuable comments.
> > We hope our additional experimental results and explanation have cleared the concern.
> > We sincerely hope the reviewer can re-evaluate our paper based on our responses, and we also welcome more comments on further improving the presentation.
> >
> > [1] Fujimoto, Scott, and Shixiang Shane Gu. "A Minimalist Approach to Offline Reinforcement Learning." arXiv preprint arXiv:2106.06860 (2021).

---

> > ### Comment · Reviewer_M22A · 2021-11-29
> > **Thanks for the detailed feedback**
> >
> > The authors' feedback addressed part of my concerns. Thus, I tend to recommend a 'weak acceptance'. I have changed my rating from 5 to 6.

---

> ### Author Response · Authors · 2021-11-26
> **Looking forward to further discussions!**
>
> Dear reviewer,
>
> We were wondering if our response and revision have resolved your concerns.
> In our responses and updated manuscripts, we gave a detailed explanation for the novelty and effectiveness of our method, especially the EVL part. Please let us know if you have any further concerns, and we are looking forward to more discussions to improve our manuscript further.
>
> Best regards,
>
> The Authors

---

> ### Author Response · Authors · 2021-11-28
> **Thank you for your time and efforts in reviewing our work !**
>
> Dear Reviewer,
>
> Thank you for your time and efforts in reviewing our work. We have provided detailed clarification and additional experiments to address the issues raised in your comments. If our response has addressed your concerns, we would be grateful if you could re-evaluate our work.
>
> If you have any additional questions or comments, we would be happy to have further discussions.
>
> Thanks,
>
> The authors

---

> ### Author Response · Authors · 2021-11-29
> **Thanks for raising the score !**
>
> We would like to thank the reviewer for raising the score!
> We really appreciate the valuable comments and suggestions from the reviewer.

---

### Official Review · Reviewer_aRTY · 2021-11-08

**Correctness:** 4
**Technical Novelty And Significance:** 3
**Empirical Novelty And Significance:** 3
**Recommendation:** 5
**Confidence:** 3

**Main Review:**

The main contribution of this paper is the introduction of the expectile operator as a smooth interpolation between behavior cloning and value learning in offline RL. Basically, the interpolation is controlled by a parameter $\tau$, so that when $\tau = 1/2$, the operator is reduced to taking expectation, and when $\tau = 1$, the operator is equivalent to the Bellman optimality operator. The authors argue that such an operator is useful in offline RL, in which case learning algorithms need to carefully balance behavior cloning and value learning to avoid extrapolation error. The authors also prove nice properties of the introduced operator, and provide empirical results to justify the effectiveness of the proposed approach.

Although this paper introduces an interesting idea, I still have the following concerns.

(i) the Bellman expectile operator is not well-defined. It seems to me that when $\tau = 1$, any value $v$ that is sufficiently large would achieve the same minimum value ($0$). In this case, how should we define the Bellman expectile operator?
(ii) The authors seem to assume that the reader is familiar with prior work on value-based offline RL. For example, in Section 2.2, $R_t$ is not defined. The algorithm in Section 3.2 is pretty hard to understand without background on episodic memory-based methods. The authors should at least give an overview on episodic memory-based methods before diving into their new methods.
(iii) It is unclear to me how one should choose $\tau$ in practice. Note that in the offline setting, one cannot simply try different $\tau$ and pick the one with the best performance, since in the offline RL setting, it is assumed that the agent does not have access to online samples. How is $\tau$ chosen in the empirical evaluation? The paper can be greatly improved the authors could give some guideline on how to pick $\tau$ in practice.

Due to the above concerns, my current recommendation is a "weak reject". However, I am open to raise my score if the authors could resolve my concerns described above.



**Summary Of The Paper:**

In this paper, the authors propose to use the expectile operator as a smooth interpolation between behavior cloning and optimal value learning in offline RL. Based on this operator, the authors develop a new offline method called Value-based Episodic Memory. The authors provide theoretical analysis and empirical results for the developed method.

**Summary Of The Review:**

See above.

---

> ### Author Response · Authors · 2021-11-21
> **Response to Reviewer aRTY**
>
> Dear Reviewer:
>
> Thanks for your inspiring and thoughtful comments. We provide clarification to your questions and concerns as below. If you have any further questions or comments, please post them, and we will be happy to have further discussions.
>
> **Q1: The definition of the Bellman expectile operator when $\tau=1$.**
>
> **A1:**
> The Bellman expectile operator is defined with $\tau\in(0,1)$.
> As the reviewer points out, when $\tau=1$, any value that is sufficiently large can be a stationary point of $\mathcal{T}\_\tau$, which is aligned with our analysis in Lemma 1, where $\tau=1$ leads to non-concentration.
> What the Bellman expectile operator guarantee is that when $\tau\rightarrow1\^-$, the stationary value $V_\tau$ approaches the optimal value $V^*$, as shown in Lemma 3.
> However, in offline RL, directly applying optimal value operator almost always leads to sub-optimal performance due to the over-generalization, so it is not sensible to set $\tau=1$ in offline RL settings.
> We thank the reviewer for pointing this out and we made this point clearer in the updated version.
>
> **Q2: The definition of $R\_t$ in Section 2.2 and an overview on episodic memory-based methods before diving into the new methods.**
>
> **A2:**
> $R\_t$ is the return augmented by expectile $V$-values, as defined recursively in Equation 7.
> We included additional details and an overview of episodic memory-based methods in Section 2 in the revision according to your suggestions.
>
> **Q3: Guidelines on how to choose $\tau$ in practice.**
>
> **A3:**
> This is an important question, and we thank the reviewer for pointing it out.
> Here we give two guidelines for choosing $\tau$ without online interactions.
>
> - **The principle for selecting $\tau$.**
> As depicted in Section 3.1, the choice of $\tau$ is mainly affected by the amount of noise in the expectile operator (defined by Equation 2) when applied to an offline dataset.
> The small amount of data, long horizon, and sparse reward environment can lead to high variance for samples in the dataset.
> So we recommend smaller $\tau$ in these scenarios to keep a conservative estimation and larger $\tau$ when you have large datasets, short-horizon tasks, or dense reward settings.
> Furthermore, we empirically find that the choice of $\tau$ is quite robust, i.e., a small change of the $\tau$ value has little effect on the performance, and thus we do not need to adjust $\tau$ very precisely.
>
> - **An approach for auto-tuning $\tau$.**
> When we have a good estimation of $V\^* $ (e.g., by utilizing some trajectories with high returns in the offline dataset), we can auto-tune $\tau$ such that the value learned by EVL is close to the estimation of $V\^* $.
> This can be done by calculating the Monte-Carlo return estimates of each state and selecting good return values as the estimation of optimal value $\tilde{V}\^* $.
> Based on this target, we develop a method for auto-tuning $\tau$, shown in Appendix A.2.
> The experimental results in Figure 9 in Appendix D.1 show that auto-tuning can lead to similar performance compared with manual selection.
>
> Thanks again for the valuable comments.
> We sincerely hope our response addresses your concern.
> We are looking forward to more discussions.

---

> ### Author Response · Authors · 2021-11-26
> **Looking forward to further discussions!**
>
> Dear reviewer,
>
> We were wondering if our response and revision have resolved your concerns.
> In our responses and updated manuscripts, we gave a more detailed explanation for our method and we included the discussion about how to choose $\tau$ in practice. Please let us know if you have any further concerns, and we are looking forward to more discussions to improve our manuscript further.
>
> Best regards,
>
> The Authors

---

> ### Author Response · Authors · 2021-11-28
> **Thank you for your time and efforts in reviewing our work !**
>
> Dear Reviewer,
>
> Thank you for your time and efforts in reviewing our work. We have provided detailed clarification and additional experiments to address the issues raised in your comments. If our response has addressed your concerns, we would be grateful if you could re-evaluate our work.
>
> If you have any additional questions or comments, we would be happy to have further discussions.
>
> Thanks,
>
> The authors

---

### Official Review · Reviewer_yPrQ · 2021-11-10

**Correctness:** 4
**Technical Novelty And Significance:** 3
**Empirical Novelty And Significance:** 3
**Recommendation:** 8
**Confidence:** 4

**Main Review:**

Strengths:
This work is well organized, the major idea is clear and valid. Besides, the proposed method is efficient and extensive experiments are provided to prove the effectiveness.

Weaknesses:
The proposed framework seems to be similar to work [1], can you compare in detail the difference and the superiority of the proposed method?

Besides, the major idea of the proposed method is also similar to the work [2]. Though work [2] is not designed for offline RL, can you also compare the major idea of the proposed method and this work?

[1]https://arxiv.org/pdf/2106.08909.pdf
[2] https://arxiv.org/pdf/2101.08152.pdf

The most recent baseline methods used are proposed in 2020, can you compare with the methods proposed in 2021, like [3]?
[3] https://arxiv.org/pdf/2106.08909.pdf


**Summary Of The Paper:**

This work proposed a new offline reinforcement learning framework, where the value function is employed to do update. Specifically, the proposed algorithm used optimal value learning and behavior cloning. Theoretical guarantees about the convergence of the proposed algorithm are provided. Besides, experiments on D4RL tasks are provided to show the effectiveness of the proposed method.

**Summary Of The Review:**

Overall, I think this is a good paper. But I am a little concerned about the novelty of this work. So I will give weak acceptance.

_____________________________________________________________

I have read the author's responses. My major concern has been solved. I decide to raise my score to 8.

---

> ### Author Response · Authors · 2021-11-21
> **Response to Reviewer yPrQ**
>
> Dear Reviewer:
>
> Thanks for your thoughtful comments. We appreciate the interesting related papers you point out, and we will give a discussion below. We hope our response can address your concern about the novelty of our method.
>
> **Q1: Compare with a recent baseline OAMPI [1].**
>
> **A1:**
> OAMPI aims to achieve a trade-off between behavior cloning and optimal value learning by restricting the number of steps for generalized policy iteration. Our method is quite different from OAMPI. The first is that we confine learning within the dataset, which serves as the core motivation of $V$-learning. Instead, multi-step OAMPI uses $\pi^i$ to sample actions that the dataset may not contain. Empirically, we find this in-distribution constraint may be insufficient compared with in-dataset constraint due to extrapolation errors. Second, our method seamlessly integrates an implicit planning module for offline learning, which greatly improves performance in sparse reward tasks like antmaze.
>
> We compare our method with OAMPI on antmaze, adriot and mujoco tasks, and the results are summarized in Table 1. In Adroit tasks, which are relatively difficult with high-dimension action space, our method significantly outperforms OAMPI by confining learning within the dataset and reducing the extrapolation error in the action space. In Antmaze tasks with sparse rewards and long horizons, our method achieves superior performance partly due to effective implicit planning. On relatively simple Mujoco tasks, our method demonstrates better or comparable performance compared with OAMPI.
>
> **Q2: Compare the major idea of the proposed method with RAPID [2].**
>
> **A2:**
> RAPID and episodic memory-based methods both consider imitating past good experiences to improve sample efficiency. However, RAPID and our methods have distinct differences. First, our method utilizes implicit planning with past experiences, which goes beyond past trajectories through combination. Instead, RAPID adds counted-based scores for selecting trajectories to improve the exploration coverage but only focuses on existing trajectories. Second, our proposed method is effective in both online and offline settings, while RAPID is not suitable for offline settings since the state space coverage is not a suitable objective in this setting.
>
> Thanks again for the valuable comments.
> We sincerely hope our response has cleared your concern regarding the novelty.
> We are looking forward to more discussions.
>
> | Type | Environments | VEM(Ours) | OAMPI |
> | ---- | ------------ | :---: | :---: |
> | fixed | antmaze-umaze | **87.5$\pm$1.1** | 82.0$\pm$2.7 |
> | play | antmaze-medium | **78.0$\pm$3.0** | 75.0$\pm$1.3 |
> | play | antmaze-large | **57.0$\pm$5.0** | 32.0$\pm$4.6 |
> | diverse | antmaze-umaze | 78.0$\pm$1.1 | **85.0$\pm$4.2** |
> | diverse | antmaze-medium | **77.0$\pm$2.2** | 70.5$\pm$3.9 |
> | diverse | antmaze-large | **58.0$\pm$2.1** | 25.0$\pm$5.2 |
> | human | adroit-door | **11.2$\pm$4.2** | 2.8$\pm$0.8 |
> | human | adroit-hammer | **3.6$\pm$1.0** | **3.9$\pm$3.0** |
> | human | adroit-relocate | **1.3$\pm$0.2** | 0.5$\pm$0.2 |
> | human | adroit-pen | **65.0$\pm$2.1** | 54.6$\pm$18.7 |
> | cloned | adroit-door | **3.6$\pm$0.3** | 0.0$\pm$0.0 |
> | cloned | adroit-hammer | **2.7$\pm$1.5** | 0.1$\pm$0.1 |
> | cloned | adroit-pen | **48.7$\pm$3.2** | 23.9$\pm$3.4 |
> | expert | adroit-door | **105.5$\pm$0.2** | 102.4$\pm$2.7 |
> | expert | adroit-hammer | **128.3$\pm$1.1** | 0.8$\pm$0.2 |
> | expert | adroit-relocate | **109.8$\pm$0.2** | 0.6$\pm$0.3 |
> | expert | adroit-pen | **111.7$\pm$2.6** | 5.1$\pm$1.8 |
> | random | mujoco-walker2d | **6.2$\pm$4.7** | 5.6 |
> | random | mujoco-hopper | **11.1$\pm$1.0** | 7.5 |
> | random | mujoco-halfcheetah | **16.4$\pm$3.6** | 3.2 |
> | medium | mujoco-walker2d | 74.0$\pm$1.2 | **81.8** |
> | medium | mujoco-hopper | 56.6$\pm$2.3 | **59.6** |
> | medium | mujoco-halfcheetah | 47.4$\pm$0.2 | **48.4** |
> | medium-replay | mujoco-walker2d | **56.6$\pm$4.2** | 49.5 |
> | medium-replay | mujoco-hopper | 74.2$\pm$6.8 | **97.5** |
> | medium-replay | mujoco-halfcheetah | 38.7$\pm$2.3 | 38.1 |
> | medium-expert | mujoco-walker2d | 110.5$\pm$5.7 | **113.0** |
> | medium-expert | mujoco-hopper | 95.8$\pm$3.7 | **103.3** |
> | medium-expert | mujoco-halfcheetah | 87.3$\pm$3.4 | **93.4** |
>
> Table 1: Comparison between VEM and OAMPI.
> The results of OAMPI in mujoco tasks are taken from [1].
> We reproduce OAMPI according to author's code: https://github.com/davidbrandfonbrener/onestep-rl.
>
> [1] Brandfonbrener, David, et al. "Offline RL Without Off-Policy Evaluation." arXiv preprint arXiv:2106.08909 (2021).
>
> [2] Zha, Daochen, et al. "Rank the Episodes: A Simple Approach for Exploration in Procedurally-Generated Environments." arXiv preprint arXiv:2101.08152 (2021).
>
> [3] Fujimoto, Scott, and Shixiang Shane Gu. "A Minimalist Approach to Offline Reinforcement Learning." arXiv preprint arXiv:2106.06860 (2021).

---

> ### Author Response · Authors · 2021-11-26
> **Looking forward to further discussions!**
>
> Dear reviewer,
>
> We were wondering if our response and revision have resolved your concerns.
> In our responses, we gave a thorough comparison with OAMPI and RAPID and made our contribution more clear. Please let us know if you have any further concerns, and we are looking forward to more discussions to improve our manuscript further.
>
> Best regards,
>
> The Authors

---

> ### Author Response · Authors · 2021-11-28
> **Thank you for your time and efforts in reviewing our work !**
>
> Dear Reviewer,
>
> Thank you for your time and efforts in reviewing our work. We have provided detailed clarification and additional experiments to address the issues raised in your comments. If our response has addressed your concerns, we would be grateful if you could re-evaluate our work.
>
> If you have any additional questions or comments, we would be happy to have further discussions.
>
> Thanks,
>
> The authors

---

> ### Author Response · Authors · 2021-11-29
> **Thanks for raising the score to 8!**
>
> We would like to thank the reviewer for raising the score!
> We really appreciate the valuable comments and suggestions from the reviewer.

---

### Official Review · Reviewer_FzBf · 2021-11-11

**Correctness:** 3
**Technical Novelty And Significance:** 2
**Empirical Novelty And Significance:** Not applicable
**Recommendation:** 8
**Confidence:** 4

**Main Review:**

## Pros
* The idea of using expectile Bellman operator to interpolate between expected Bellman operator and Bellman optimality operator is interesting.
* The authors has conducted experiments in more than 20 environments and the proposed method's performance either surpasses SOTA methods or on-par with them.

-----
## Cons/questions/suggestions [C/Q/S]
* [C1]  The clarity of the manuscript could be significantly improved as some of the discussions/explanations are either not precise or consistent with each other. __Example 1__ On page 3, the authors wrote "The value estimation learned by EVL, on the contrary, achieves a trade-off between learning optimal policy and behavior cloning and can be close to the optimal value with proper chosen ... The estimation error can be significant when the dataset is small, and EVL needs a smaller to be more conservative and closer to behavior". When the dataset is small, I understand that one want to decrease the $\tau$ to be more conservative. But it's not clear to me whether the smaller the better in this case. Also, when the authors say EVCL needs to be closer to behavior cloning, does that mean we should just choose $\tau = 0.5$ in this case? __Example 2__ The optimal values for $\lambda$ are mostly below 0.5 as seen in table 3. Based on the reasoning given by authors, this seems to indicate that these datasets used in the experiments are "small" because we require small $\lambda$ to obtain good performance. However, the datasets in D4RL are quite large and often much bigger than most real datasets. I feel these results are inconsistent with the authors' discussion in the aforementioned example . __Example 3__ On page 3, the authors wrote " VEM uses expectile V -learning (EVL) to learn V -functions while avoiding extrapolation error in the action space." This claim seems to be too strong for me. As "avoid" indicates completely removing the extrapolation error, while EVL actually just serves to reduce the extrapolation error. I think this claim should be more precise.  __Example 4__. On page 3, the authors wrote "in real-world problems, the dynamics are often nearly deterministic". For robotics applications, this might be true. However, for a lot of other real-life applications, for example, healthcare, industrial control, autonomous driving, recommender systems, etc, I don't think the dynamics is anywhere near deterministic.
* [C2] The authors emphasize that EVL could fundamentally avoid unseen actions. I don't see why this is the case. I feel the policy network learned through advantage-weighted loss (eqn.7) can defintiley give out-of-distribution actions when taking the argmax due to inaccurate advantage estimations. Please clarify on this.
* [Q1] In Figure 10 (b - e), the value estimation error never decreased, and the optimal estimations were obtained at 0th step. This seems to indicate that the value networks failed at learning in the corresponding environments. However, in table 1, good performances are reported for these environments. Could the authors explain why this is the case?
* [Q2] AWR is a value-based offline RL method and is probably the most relevant baseline. I'm wondering whether the authors could provide some explanations on why AWR failed at four out of six dataset types for antmaze while VEM performs well for all six.
* [S1] For equation 2, the authors mention that when $\tau\rightarrow 1$, the Bellman expectile operator approaches the Bellman optimality operator. However, the lemma 3 for this was not mentioned until page 5. I suggest adding a sentence referring readers to lemma 3 for this important observation.
* [S2] When $\lambda=0.5$, VEM is essentially behavior cloning + implicit planning. I think this baseline should be listed separately in table 1 to help readers see the importance of introduced flexibility for value learning with the Bellman expectile operator.

----
## Minor comments
* [Typo] Page 8, "Therefore, the generative model in BCQ cannot __guarantees__ completely" -> guarantee
* [Plot] Figure 4 is quite hard to read. The authors could probably just get rid of the floor pattern and make it a different color that has higher contrast with the value pixels.

**Summary Of The Paper:**

The authors proposed a new offline RL method called VEM that combines value expectile learning and implicit planning, with the goal of reducing extrapolation error in the action space. The proposed value expectile learning is an interpolation between Bellman expectation operator and Bellman optimality operator. By adjusting the hyperparemeter, one can adjust the tradeoff between sticking to behavior policy or deviate from it to potentially obtain better performance. On standard offline RL benchmarks, the authors demonstrate that the proposed method achieve SOTA performance.

**Summary Of The Review:**

Though the idea of balancing between behavior cloning and optimal value learning through the Bellman expectile operator is interesting, the clarity of the current manuscript still require some improvement. If the authors could address my concerns and questions, I'd be happy to adjust the score.

---

> ### Author Response · Authors · 2021-11-21
> **Response to Reviewer FzBf (Part I)**
>
> Dear Reviewer,
>
> We thank the reviewer for the detailed and insightful comments. We add supplementary experimental results as part of our response and clarify the reviewer's questions and concerns as follows.
>
> **C1: The clarity of the manuscript.**
>
> **A1**:
> We first would like to clarify that $\tau=0.5$ does not necessarily mean behavior cloning since VEM uses implicit memory-based planning. As pointed out in [1], the planning is essentially taking the maximum over the $n$-step returns, leading to potential overestimation due to approximation errors and the nature of the maximum operator. This requires additional conservatism to reduce the potential overestimation.
>
> **C1-E1: Whether the smaller, the better for $\tau$ when the dataset is small? And whether we should just choose $\tau=0.5$ in this case?**
>
> **A-C1-E1**: For a given small dataset, smaller $\tau$ does not necessarily lead to better performance. We try to deliver in this paragraph precisely what the reviewer agrees with; that is, as the size of the dataset decreases, we may choose a proper smaller $\tau$ to be more conservative. By doing so, EVL can accommodate variant sizes of datasets. The choice of $\tau$ is also affected by other factors like the quality of the dataset. We thank the reviewer for pointing this out, and we made this point clearer in the updated version of the submission.
>
> **C1-E2: Why are the optimal values for $\tau$ mostly below 0.5 even in large datasets in D4RL?**
>
> **A-C1-E2**: As mentioned in **A1**, choosing a smaller $\tau$ is because of the overestimation nature of implicit memory-based planning. As shown in Equation 9, the implicit memory-based planning can take the maximum over the $n$-step returns.
> We conduct an additional ablation study to demonstrate it.
> The experimental results in Figure 10 in Appendix D.1 in the updated paper show the overestimation error introduced by the memory-based planning.
>
> **C1-E3: The claim that EVL can avoid extrapolation error in the action space is too strong. Whether policy network can definitely give out-of-distribution actions when taking the argmax due to inaccurate advantage estimations?**
>
> **A-C1-E3**: For this claim, we refer to the fact that EVL does not require any actions outside of the dataset in the **value learning** phase. Regarding the policy learning phase, our method needs to generalize to unseen actions where extrapolation error is unavoidable, as the reviewer mentioned. In the value learning phase, we empirically find it can be insufficient to confine actions within a small range from samples in the dataset, and confining the value learning completely *within the dataset* is beneficial.
>
>
> **C1-E4: The claim that the dynamics in real-world are often nearly deterministic is not suitable.**
>
> **A-C1-E4**:
> We agree that there are real-world applications, such as recommendation systems, that can have high stochasticity. In EVL, we focus on settings where the environment is nearly deterministic, such as robotics.
>
> **C2: The claim that EVL could avoid unseen actions.**
>
> **A-C2**:
> As explained in **A-C1-E**, for this claim, we refer to the fact that EVL does not require any actions out of the dataset in **value learning** phase, while many other methods require a policy network in this phase, which may lead to inaccurate value estimation.
>
> Overall, we thank the reviewer again for the detailed comments, and we made these points more clear and explicit in the updated version.

---

> > ### Author Response · Authors · 2021-11-21
> > **Response to Reviewer FzBf (Part II)**
> >
> > **Q1: In Figure12(b-e), the value estimation error never decreased, and the optimal estimations were obtained at 0-th step.**
> >
> > **A-Q1**:
> > Figure 12 shows the estimation error of the current value evaluation from the Monte-Carlo return of the current policy rather than from the optimal value. Figure12 (b-e) belongs to the antmaze tasks, where the reward is 1 if the ant reaches a goal and, otherwise, the reward is 0. For this reason, the value estimates are just the true values in the online evaluation since the initialized policy has a zero return. The estimation error is restricted to very low values (less than 0.5) during the training process rather than divergent. In other tasks, the estimation error has a clear tendency to decrease. Therefore, the value network has reasonable estimations.
> >
> > **Q2: Why AWR failed at four out of six dataset types for antmaze while VEM performs well for all six?**
> >
> > **A-Q2**:
> > There are two reasons for the superiority of VEM over AWR: expectile value learning (EVL) and episodic-memory (EM) implicit planning. To illustrate this point, we conducted an ablation study on VEM and compared them with AWR. Since AWR is a special case of VEM without EVL and EM, our ablation study includes two additional baselines: VEM-EVL (VEM without EVL) and VEM-EM (VEM without memory-based implicit planning). The experimental results show that memory-based implicit planning greatly contributes to VEM's performance and EVL provides an additional trade-off. EM can extend the planning horizon and is effective for tasks like antmaze, which have sparse rewards. Also, for a more fair comparison, we add results of our implementation of AWR in addition to results from [2].
> >
> > | Type | Env    | VEM | VEM - EVL    | VEM - EM     | AWR  | AWR(Ours)    |
> > |-----|--------|--------|-------|-----|------|--------------|
> > | fixed  | umaze  | 87.5$\pm$1.1     | 85.0$\pm$1.5 | 74.0$\pm$3.0 | 56.0 | 58.8$\pm$2.5 |
> > | play  | medium | 78.0$\pm$3.1   | 71.0$\pm$2.5 | 64.0$\pm$3.7 | 0.0  | 58.7$\pm$3.5 |
> > | play  | large  |  57.0$\pm$5.0   | 45.0$\pm$2.5 | 26.0$\pm$1.5 | 0.0  | 21.1$\pm$2.5 |
> > | diverse | umaze  |  78.0 $\pm$ 1.1 | 75.0$\pm$5.0 | 70.0$\pm$2.5 | 70.3 | 67.7$\pm$3.0 |
> > | diverse | medium |  77.0$\pm$2.2   | 60.0$\pm$5.0 | 43.0$\pm$4.1 | 0.0  | 22.2$\pm$3.5 |
> > | diverse | large  |  58.0 $\pm$ 2.1 | 48.0$\pm$2.7 | 41.0$\pm$3.5 | 0.0  | 20.0$\pm$2.5 |
> >
> > Table 1. Comparison between VEM, VEM-EVL, VEM-EM and AWR on antmaze tasks.
> > The results for AWR are taken from [1].
> > The results for AWR (Ours) are from our re-implementation of AWR for antmaze tasks.
> >
> > **S1: The lack of timely explanation for the Bellman expectile operator.**
> >
> > **A-S1**: We thank the reviewer for pointing this out and we added some explanation in the updated version.
> >
> > **S2: Behavior cloning ($\tau$=0.5) + implicit planning should be listed separately in Table 1 as a baseline.**
> >
> > **A-S2**:
> > We have added the result of VEM($\tau$=0.5) in Table 1 in the updated version.
> > The significant performance improvement in antmaze and adroit/cloned tasks illustrates the importance of the Bellman expectile operator due to the flexibility for value learning.
> > Note that when $\tau=0.5$, VEM is *not* behavior cloning + implicit planning based on the analysis in **A-C1-E2**.
> > Nevertheless, VEM($\tau=0.5$) is still a good baseline to show the importance of the trade-off provided by the expectile operator.
> >
> > **Minor comments**: Typo error and the Figure 4 is quite hard to read.
> >
> > **A-Minor comments**: We thank the reviewer for the detailed comment, and we corrected this in the updated paper according to your suggestions.
> >
> > Thanks again for your supportive comments and detailed instructions.
> > We sincerely hope that our response has cleared your concerns and questions regarding the clarity and experimental results.
> >
> > [1] Hu, Hao, et al. "Generalizable episodic memory for deep reinforcement learning." arXiv preprint arXiv:2103.06469 (2021).
> > [2] Fu, Justin, et al. "D4rl: Datasets for deep data-driven reinforcement learning." arXiv preprint arXiv:2004.07219 (2020).

---

> > > ### Comment · Reviewer_FzBf · 2021-11-29
> > > **.**
> > >
> > > I'd like to thank the authors for their detailed response and for incorporating additional results. My questions and concerns are addressed. And these new results greatly strengthen the quality of the paper. Thus, I would like to recommend acceptance of this manuscript and I have updated my rating from 5 to 8.

---

> ### Author Response · Authors · 2021-11-26
> **Looking forward to further discussions!**
>
> Dear reviewer,
>
> We were wondering if our response and revision have resolved your concerns.
> In our responses, we have made our manuscript more clear and included more experimental results to demonstrate the effectiveness of our method. Please let us know if you have any further concerns, and we are looking forward to more discussions to improve our manuscript further.
>
> Best regards,
>
> The Authors

---

> ### Author Response · Authors · 2021-11-28
> **Thank you for your time and efforts in reviewing our work !**
>
> Dear Reviewer,
>
> Thank you for your time and efforts in reviewing our work. We have provided detailed clarification and additional experiments to address the issues raised in your comments. If our response has addressed your concerns, we would be grateful if you could re-evaluate our work.
>
> If you have any additional questions or comments, we would be happy to have further discussions.
>
> Thanks,
>
> The authors

---

> ### Author Response · Authors · 2021-11-29
> **Thanks for raising the score to 8!**
>
> We would like to thank the reviewer for raising the score!
> We really appreciate the valuable comments and suggestions from the reviewer.

---

### Author Response · Authors · 2021-11-26
**General Remark**

Dear Reviewers,

We first thank the reviewers for their insightful and positive feedback.
We are encouraged that the reviewers found our work "interesting" (FzBf, aRTY), "theoretical analysis convincing" (M22A, 3Tyw, y86L), "experimental results extensive" (M22A, yPrQ).
In terms of writing, we are also pleased the reviewers found our paper is clear (yPrQ) and easy to understand (y86L).

In our response, we have provided additional experimental results and improved the clarity of the manuscript. Specifically, we compare VEM with several important baselines (e.g., OAMPI [1], TD3+BC [2], MOReL [3] and COMBO [4]) on the full set of D4RL mujoco/adroit/antmaze/kitchen datasets in supplementary experimental results.
We also add more detailed explanations for our algorithm and make our contribution more clear.
We sincerely hope that our response can clear reviewers' concerns.
More discussions and suggestions on further improving the paper are also always welcomed!

[1] Brandfonbrener, David, et al. "Offline RL Without Off-Policy Evaluation." arXiv preprint arXiv:2106.08909 (2021).

[2] Fujimoto, Scott, and Shixiang Shane Gu. "A Minimalist Approach to Offline Reinforcement Learning." arXiv preprint arXiv:2106.06860 (2021).

[3] MOReL: Model-Based Offline Reinforcement Learning.  Kidami et al.,2020

[4] COMBO: Conservative Offline Model-Based Policy Optimization. Yu et al., 2021.

---

### Decision · Program_Chairs · 2022-01-20

**Decision:**

Accept (Poster)

**Comment:**

Most of the reviewers think this paper is clearly a valuable addition to ICLR based on the convincing theoretical analysis and extensive experimental results. Please refer to reviewers's review for more detailed discussions of the pros and cons of the paper.